# Proteomic screens of SEL1L-HRD1 ER-associated degradation substrates reveal its role in glycosylphosphatidylinositol-anchored protein biogenesis

Xiaoqiong Wei [1,2,7], You Lu[2,3,7], Liangguang Leo Lin[1,2,7], Chengxin Zhang [4], Xinxin Chen[1,2], Siwen Wang[2], Shuangcheng Alivia Wu[1,2], Zexin Jason Li[1,5], Yujun Quan[1], Shengyi Sun [6] & Ling Qi [1,2] ✉

Endoplasmic reticulum-associated degradation (ERAD) plays indispensable roles in many physiological processes; however, the nature of endogenous substrates remains largely elusive. Here we report a proteomics strategy based on the intrinsic property of the SEL1L-HRD1 ERAD complex to identify endogenous ERAD substrates both in vitro and in vivo. Following stringent filtering using a machine learning algorithm, over 100 high-confidence potential substrates are identified in human HEK293T and mouse brown adipose tissue, among which ~88% are cell type-specific. One of the top shared hits is the catalytic subunit of the glycosylphosphatidylinositol (GPI)-transamidase complex, PIGK. Indeed, SEL1L-HRD1 ERAD attenuates the biogenesis of GPI-anchored proteins by specifically targeting PIGK for proteasomal degradation. Lastly, several PIGK disease variants in inherited GPI deficiency disorders are also SEL1L-HRD1 ERAD substrates. This study provides a platform and resources for future effort to identify proteome-wide endogenous substrates in vivo, and implicates SEL1L-HRD1 ERAD in many cellular processes including the biogenesis of GPI-anchored proteins.

Endoplasmic reticulum-associated degradation (ERAD), a principal protein quality-control mechanism in the ER, plays an essential role in regulating protein abundance and biogenesis in the ER and the crosstalk among organelles[1–3]. Among several ERAD protein complexes, the SEL1L–HRD1 (Hrd3–Hrd1 in yeast) complex represents the most conserved branch of ERAD. SEL1L is not only a scaffolding protein for the ERAD complex and HRD1 stability[4–8] but also recruits substrates through the ER-resident lectins OS9 and ERLEC1 (XTP3-B)[9–11]. Recent studies using cell type-specific mouse models including those from our laboratory have demonstrated the vital importance of SEL1L and HRD1 in a cell type- and substrate-specific manner in physiology[2,12–35]. Moreover, we recently reported the identification of 11 patients carrying SEL1L and HRD1 variants with ERAD-associated neurodevelopmental disorders with onset infancy (ENDI) syndrome[36,37]. Despite these recent advances in elucidating its pathophysiological importance, our understanding of this complex remains limited as the

[1]Department of Molecular Physiology and Biological Physics, University of Virginia, School of Medicine, Charlottesville, VA 22903, USA. [2]Department of Molecular and Integrative Physiology, University of Michigan Medical School, Ann Arbor, MI 48105, USA. [3]Life Sciences Institute and Department of Cell & Developmental Biology, University of Michigan Medical School, Ann Arbor, MI 48109, USA. [4]Department of Computational Medicine and Bioinformatics, University of Michigan Medical School, Ann Arbor, MI 48105, USA. [5]Department of Biological Chemistry, University of Michigan Medical School, Ann Arbor, MI 48105, USA. [6]Department of Pharmacology, University of Virginia, School of Medicine, Charlottesville, VA 22903, USA. [7]These authors contributed equally: Xiaoqiong Wei, You Lu, Liangguang Leo Lin. ✉e-mail: xvr2hm@virginia.edu

number and nature of its endogenous substrates, especially in vivo, remain largely unknown.

Previous attempts to identify proteome-wide substrates by blocking, either chemically[38] or genetically[39–41], of the ERAD pathway, have generated a few hits—likely those with high abundance or high propensity to misfold. This was largely due to the transient nature of the interactions between the ERAD complex and misfolded protein substrates. For example, a SILAC-based screening based on elevated abundance in *HRD1* knockout (KO) vs. wildtype (WT) HEK293T cells identified only 56 potential candidate substrates, most of which are components of highly abundant ERAD machinery such as OS9, BiP/HSPA5 and UBE2J1, and ER chaperones[40]. Tyler et al.[41] identified 14 OS9-interacting proteins in *SEL1L KO* vs. WT HEK293T cells, most of which were ERAD cofactors and ER chaperones. These results highlight an urgent need for a new, more sensitive, approach to identify endogenous substrates. As the importance of SEL1L–HRD1 ERAD in health and disease is emerging[35–37], gaining a comprehensive picture of endogenous substrates in vivo will not only be a key to uncover the pathophysiological significance of SEL1L–HRD1 ERAD and its underlying molecular mechanism but also be important for future therapeutic strategies to target this protein complex.

Many ER luminal proteins lacking transmembrane domains are anchored to the membranes via glycosylphosphatidylinositol (GPI), a post-translational modification with a glycolipid[42]. In humans, there are over 150 GPI-anchored proteins with a wide array of functions, including enzymes, adhesion molecules, receptors, protease inhibitors, transporters, and complement regulatory proteins[43]. Attachment of pre-assembled GPI to the protein moiety is mediated by the ER-resident GPI transamidase complex, composed of five subunits phosphatidylinositol glycan anchor biosynthesis class K (PIGK), S (PIGS), T (PIGT), U (PIGU), and glycosylphosphatidylinositol anchor attachment 1 (GPAA1). PIGK cleaves GPI signal peptides and generates a substrate–enzyme complex via a thioester bond[44]. PIGK forms an intermolecular disulfide bond with PIGT, which stabilizes PIGK and is required for its assembly into the GPI transamidase complex[45,46]. Importantly, many genetic variants have been identified in components of this complex in patients with neurodevelopmental disorders[47–53], termed inherited GPI deficiency disorders (IGDs). However, how the biogenesis and maturation of nascent GPI transamidase complex, both WT and disease mutants, take place in the ER remains largely unknown.

Here we have developed an immunoprecipitation (IP)-based proteomic screening strategy to identify endogenous ERAD substrates both in vitro (HEK293T cells) and in vivo (brown adipose tissue, BAT). The principle behind this screen is that SEL1L–substrate interactions are prolonged and stabilized in the absence of E3 ligase HRD1, which can be recognized by a high-affinity SEL1L antibody[32]. By applying a machine learning algorithm, over 100 hits are identified as potential endogenous substrates in each cell type including many membrane and luminal proteins involved in many cellular processes. This list is available on our server [https://lingqiserver.github.io/ipmassspec]. Following the validation of several hits, we further show that SEL1L–HRD1 ERAD attenuates the biogenesis of GPI-anchored proteins, at least in part, by targeting PIGK protein for proteasomal degradation. Providing the clinical relevance of these findings, we further show that several PIGK disease mutants are SEL1L–HRD1 ERAD substrates, which prevents them from forming high molecular-weight protein aggregates.

## Results

### Development of SEL1L IP-based proteomic screen

In SEL1L–HRD1 ERAD, SEL1L is involved in substrate recruitment and HRD1 stability[5,7,54–56]. The interaction between SEL1L and a model substrate HA-tagged ProAVP^G57S, a previously characterized ERAD substrate[14,37], was barely detectable in WT HEK293T cells following HA-IP (lane 6, Fig. 1a), in line with the transient and unstable nature of SEL1L–substrate interaction. However, loss of HRD1 stabilized the interaction between SEL1L and the substrates in human kidney cell line HEK293T cells (lane 8 vs. 6–7, Fig. 1a).

To leverage this intrinsic property of the SEL1L–HRD1 ERAD complex to identify proteome-wide endogenous ERAD substrates, we generated several SEL1L-specific antibodies and vigorously verified their efficacy in IP. One of them was suitable for IP of endogenous SEL1L protein in mammalian cells (Fig. 1b). The antibody efficiently

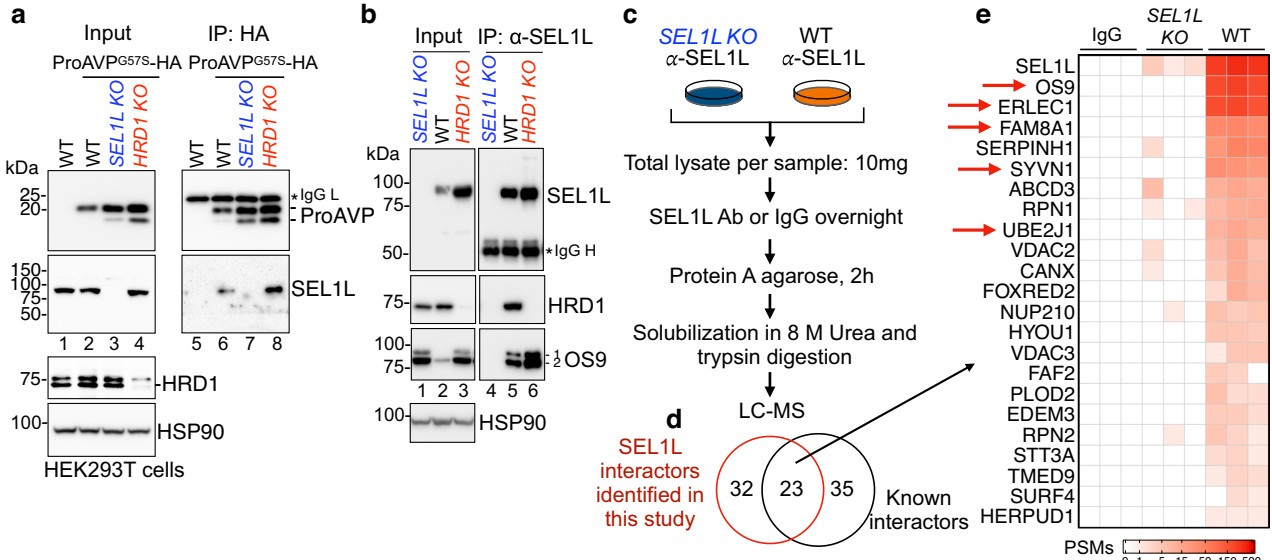

**Fig. 1 | Development of a SEL1L IP-based proteomic screen strategy. a** Immunoprecipitation of HA-ProAVP^G57S in HEK293T cells transfected with a known ERAD substrate ProAVP^G57S-HA, showing that the association between the substrate and SEL1L was stabilized in *HRD1 KO* cells (n = 3 independent repeats). Asterisk indicates a non-specific cross-reactive band. **b** Immunoprecipitation using anti-SEL1L antibody in HEK293T cells showing the efficient immunoprecipitation of endogenous ERAD components HRD1 and OS9 (n = 2 independent repeats). **c** Flow chart of anti-SEL1L IP-MS to detect endogenous SEL1L-interacting proteins in HEK293T. **d** Comparative analysis with previously reported interactors of the SEL1L–HRD1–ERAD complex. **e** Heatmap of SEL1L interactors. Plotted values are peptide-spectrum matches (PSMs) for each protein in each sample from three independent experiments. Source data are provided as a Source Data file.

immunoprecipitated known endogenous SEL1L–HRD1 ERAD complex including HRD1 and OS9 proteins in WT, but not *SEL1L KO*, HEK293T cells (lane 5 vs. 4, Fig. 1b). Indeed, quantitative liquid chromatography–tandem mass spectrometry (LC–MS/MS) analyses of endogenous SEL1L interactome in WT vs. *SEL1L KO* HEK293T cells (Fig. 1c) identified 55 hits, almost half of which were known to interact with SEL1L as previously reported[8,41,57] (Fig. 1d). Most of the known components of the SEL1L–HRD1 ERAD complex, including HRD1 (SYVN1), OS9, ERLEC1 (XTP3-B), FAM8A1 and UBE2J1, were indeed identified as the top hits (Fig. 1e). Hence, we have successfully developed a system applicable to pull down the whole SEL1L–HRD1 complex.

### Identification of high-confidence endogenous SEL1L–HRD1 ERAD substrates

Using this system, we next went on to identify endogenous substrates from *HRD1 KO* samples compared to WT and *SEL1L KO* samples (Fig. 2a). To gain a glimpse of cell type-specific ERAD substrates both in vitro and in vivo, we performed the experiments in both human HEK293T cells and mouse BAT. We also developed a streamlined data analysis pipeline that selected protein hits enriched in *HRD1 KO* cells relative to WT to a higher degree than changes in mRNA levels measured by transcriptomics RNA-seq analyses, followed by a data-driven machine learning algorithm that generated a confidence score for unbiased substrate selection (Supplementary Fig. 1a). This machine learning model was trained on proteins previously confirmed to be ERAD substrates as positive hits, and proteins determined to be non-substrates based on the subcellular localization, post-translational modifications, transmembrane regions, and signal peptides (or lack thereof) as negative hits. Input features of the model were derived from peptide-spectrum matches (PSMs) of the proteins in different samples and fed into a logistic regression model, which was then optimized by gradient descent to maximize the prediction accuracy of substrate versus non-substrate on the training set (see the "Methods" section for details).

A total of 2923 unique protein hits were identified from three independent SEL1L-IP-MS experiments. To exclude candidates enriched in *HRD1 KO* vs. WT HEK293T cells due to transcriptional up-regulation, we measured their transcriptomes using RNA-seq (Supplementary Fig. 1b). There were near-zero Pearson correlation coefficients when comparing changes in PSMs vs. those in the mRNA levels between *HRD1 KO* and WT HEK293T cells, suggesting the enrichment of SEL1L-immunoprecipitants in *HRD1 KO* cells was largely independent of transcriptional upregulation (Supplementary Fig. 1c). Upon selecting the hits with high confidence scores and excluding those with significant upregulation in mRNA levels, a total of 119 positive hits were identified in at least two independent repeats (Fig. 2b and Supplementary Data 1). The PSMs distribution for these hits were highly consistent between the experiments with high Pearson correlation coefficients (Supplementary Fig. 1d). We classified these hits into two groups based on whether they were detected in WT samples or not (Fig. 2c): Group A contained 47 proteins that were detected in WT samples and further accumulated in *HRD1 KO* samples, including several known ERAD substrates such as major histocompatibility class I (MHC-I) heavy chain molecules HLA-C[58], unfolded protein response (UPR) sensor ATF6[59] and ERAD components OS9[40] (Fig. 2d and Supplementary Fig. 2a). Group B contained 72 proteins which were only detectable in *HRD1 KO* cells, but not in WT cells, representing those whose interaction with SEL1L–HRD1 ERAD is very transient or protein level is relatively low in WT HEK293T cells (Fig. 2e and Supplementary Fig. 2b). Both MHC-I, including HLA-E and HLA-F, and mitochondria-associated membranes protein SigmaR1, a previously reported endogenous SEL1L–HRD1 ERAD substrate[34,58], were in this group. Most of the candidates were new, with 12 and 4 hits identified in previous

studies[40,41], including OS9 and a cell adhesion protein CERCAM (Fig. 2f, g). These findings not only validated our system but also highlighted the sensitivity of our screening methods.

61% of the putative SEL1L–HRD1 substrates were membrane proteins with the rest being luminal soluble proteins, 69% with glycosylation, and 31% with disulfide bonds (Fig. 2d, e and Supplementary Fig. 2a, b). Moreover, pathway analyses showed that putative substrates are involved in a wide array of cellular processes, from many occurring in the ER (e.g. protein glycosylation, protein folding, protein targeting to ER, UPR, antigen presentation, regulation of immune response, and GPI-anchored protein biosynthesis) to processes associated with other organelles or extracellular matrix (e.g. mitochondrial function, lysosomal function, nuclear transport, glycosaminoglycan metabolism, transmembrane transport, collagen biosynthesis, lipid metabolism, and cell adhesion) (Fig. 2h).

We next took a step further to ask whether this label-free strategy is applicable in vivo. To this end, we performed SEL1L IP-MS using BAT from *Sel1L^{Ucp1Cre}* or *Hrd1^{Ucp1Cre}* mice[34], as well as differentiated *Sel1L KO* and *Hrd1 KO* white adipocytes generated using the CRISPR/Cas9 technology. We identified a total of 152 proteins enriched in *Hrd1*-deficient BAT/adipocytes relative to their WT samples in at least two experiments (Fig. 3a; Supplementary Data 2). Consistent with the findings in HEK293T cells, putative ERAD substrates enriched in *HRD1 KO* over WT cells were largely uncoupled from changes in mRNA levels (Supplementary Fig. 3a, b). Similarly, we divided substrates into two groups based on whether the interaction was detected in WT samples: Group A with 18 hits that were detected in WT samples and Group B with 134 candidates that were not (Fig. 3b–d, Supplementary Fig. 3c). ~60% of the hits were, similar to those in HEK293T cells, transmembrane proteins with ~45% being glycosylated and ~30% with disulfide bonds. Group A included OS9, UBE2J1, MHC class I (H2-K1), and lipoprotein lipase (LPL)[32,33], while Group B included calnexin (CANX) (Fig. 3c, d). Functional categorization revealed that these candidates are associated with many cellular processes, including processes occurring in the ER, mitochondrial function, transmembrane transport, collagen biosynthesis, lipid metabolism, and cell adhesion (Fig. 3e). One noticeable difference with the HEK293T dataset was that a large number of the candidates was mitochondrial proteins involved in mitochondrial organization (AarF domain-containing protein kinase ADCK1), mitochondrial transport (SLC25A1, 16 and 35 proteins) or respiratory electron transport chain (NADH dehydrogenase NDUFA1, 9 and 13 proteins) (Fig. 3e).

### Shared vs. cell-type-specific ERAD substrates

We next performed comparative analyses between the two datasets to identify cell type-specific vs. common substrates (Fig. 4a; Supplementary Data 3). Among the total 238 hits identified in both cell types, 29 (25% for HEK293T and 19% for BAT) were shared between the two cell types (Fig. 4a, b). The shared pathways included diverse cellular processes, such as ER protein folding and processing, cell adhesion and migration, GPI-anchored protein biosynthesis, cell cycle and differentiation, collagen biosynthesis, and transmembrane transport (gray, Fig. 4e). The pathways enriched among shared hits included proteins involved in ERAD, protein glycosylation, immune responses and protein targeting to the ER (yellow, Fig. 4e), pointing to a possible close relationship between ERAD and these cellular processes, esp. protein glycosylation.

About 87 (75% for HEK293T) and 122 (81% for BAT) hits were cell type-specific (Fig. 4a, c–e; Supplementary Data 3). In line with the thermogenic function of brown adipocytes, substrates involved in mitochondrial function, lipid metabolism, redox regulation, and secretory pathway were highly enriched in BAT (Fig. 4d and pink, Fig. 4e). By contrast, proteins involved in lysosomal function, glycosaminoglycan biosynthesis, nuclear transport and UPR were highly represented in HEK293T cells (Fig. 4c and blue, Fig. 4e). Cell type-

specific expression of a subset of these hits based on the mRNA levels such as adiponectin (ADIPOQ) in BAT (purple circles, Fig. 4c, d) could explain cell type-specific effect of ERAD, but not for the other proteins that are expressed in both cell types. These findings highlight the importance of identifying cell-type-specific SEL1L–HRD1 ERAD substrates and provide strong support for its cell-type-specific effect in vivo as recently reported[2,3,35].

## Validation of putative ERAD substrates

We next validated a subset of the hits, including two shared substrates, Group XIIA secretory phospholipase A2 (PLA2G12A) and ER chaperone Malectin (MLEC), and cell type-specific substrates, alpha-L-fucosidase (FUCA2), ADIPOQ and LPL. PLA2G12A is a secreted phospholipase, with no predicted glycosylation and disulfide bond[60], while MLEC involved in protein glycosylation is a transmembrane ER-resident glycoprotein

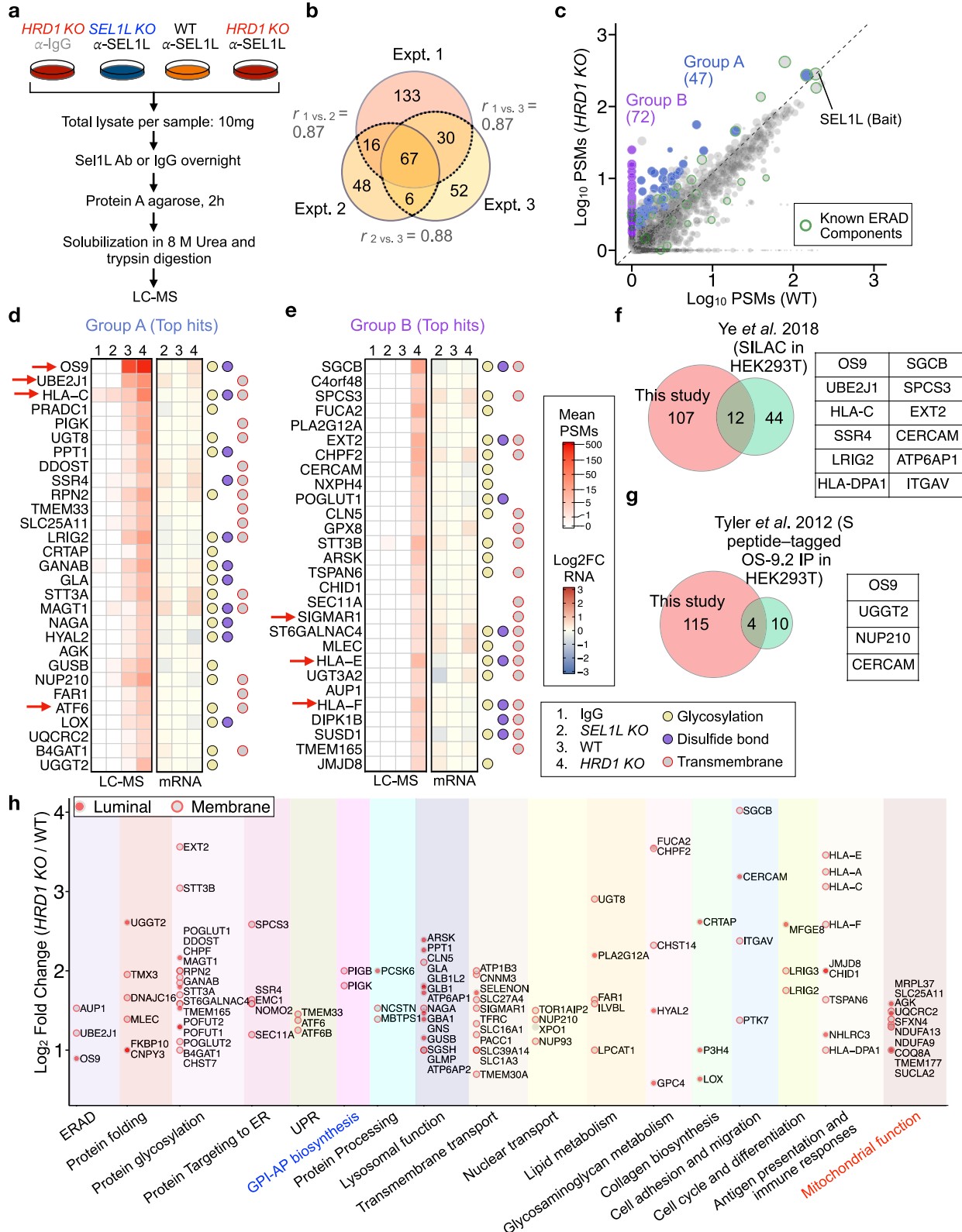

**Fig. 2 | Proteomic screening for SEL1L–HRD1 ERAD substrates in HEK293T cells.** **a** Flow chart of the SEL1L-IP-MS strategy to screen for endogenous ERAD substrates in HEK293T cells. **b** Venn diagram showing the overlaps of substrate candidates identified from three independent experiments. Pearson correlation coefficients (*r*) were calculated using the PSMs from *HRD1* samples for each overlapping candidate. **c** Scatter diagram showing peptide-spectrum matches (PSMs) in WT and *HRD1 KO* samples for each non-redundant protein hit from SEL1L-IP-MS in HEK293T. The slope of the dashed black line denotes the cutoff value calculated from PSMs of the bait (SEL1L) in *HRD1 KO* compared to WT samples used for substrate selection. ERAD substrate candidates detectable or non-detectable in WT samples are classified as Group A (in blue) and B (in purple). Dot size is proportional to the protein's mean score from three experiments. Known ERAD pathway component proteins are highlighted with green circles. **d** and **e** Heatmaps showing mean PSMs from IgG, *SEL1L KO*, WT, and *HRD1 KO* SEL1L-IP-MS samples and RNA log2 fold change (FC) in *KO* relative to WT samples for the top ERAD substrate candidates in Group A (**d**) and Group B (**e**) ranked by mean weighted scores from three experiments. Dot plots on the right indicate the presence of protein N-glycosylation, disulfide bonds, and transmembrane domains. **f** and **g** Venn diagrams showing the overlap between putative ERAD substrates identified in this study and the HEK293T SILAC proteomics dataset published in Ye et al. 2018[40] (**f**) and the S peptide-tagged OS9.2 IP-MS in HEK293T published in Tyler et al. 2012[41] (**g**). Substrate candidates in common are listed to the right. **h** Functional categorization and mean fold enrichment in *HRD1 KO* over WT samples for each ERAD substrate candidate. Luminal or membrane topology is annotated as filled red dots or open red circles, respectively, for both ER and mitochondrial proteins. GPI-AP, Glycosylphosphatidylinositol-anchored protein.

with one transmembrane domain and no predicted disulfide bond[61,62] (Fig. 5a). PLA2G12A and MLEC were accumulated in both HEK293T cells and BAT when SEL1L or HRD1 were deficient (Fig. 5b), in a transcription-independent manner (Supplementary Fig. 4a). Degradation of PLA2G12A and MLEC in HEK293T cells was indeed SEL1L- and HRD1-dependent as both proteins were significantly stabilized in the absence of SEL1L–HRD1 ERAD (Fig. 5c, d).

FUCA2 is a secreted enzyme with three N-glycosylation sites (Fig. 5e), which has been implicated as a prognostic biomarker for multiple cancer types[63] and is involved in *Helicobacter pylori* adhesion to human gastric epithelial cells[64]. FUCA2 was accumulated at much higher levels in ERAD-deficient HEK293T cells compared to WT cells (Fig. 5f), uncoupled from gene transcription (Supplementary Fig. 4b). This was specific for HEK293T cells, as FUCA2 protein level was unchanged in ERAD-deficient BAT (Fig. 5f). Degradation of FUCA2 in HEK293T cells was indeed SEL1L- and HRD1-dependent as FUCA2 was significantly stabilized in the absence of SEL1L–HRD1 ERAD (Fig. 5g-h).

LPL and ADIPOQ are two abundant secreted proteins specifically expressed in adipocytes with important metabolic functions[65,66]. LPL contains multiple disulfide bonds and two N-glycosylation sites, while ADIPOQ only has one intermolecular disulfide (Fig. 5i). In line with a recently published study[32], LPL was indeed an adipocytes-specific ERAD substrate and accumulated in ERAD-deficient BAT (Fig. 5j). Similarly, ADIPOQ accumulated in BAT when SEL1L–HRD1 ERAD was abolished (Fig. 5k). Accumulation of both LPL and ADIPOQ in ERAD-deficient BAT was uncoupled from gene transcription (Supplementary Fig. 4c). Hence, we concluded that PLA2G12A, MLEC, FUCA2, ADIPOQ, and LPL are bona fide endogenous SEL1L–HRD1 ERAD substrates, the first two being shared substrates and the latter three being cell type-specific substrates.

## PIGK is an endogenous SEL1L–HRD1 ERAD substrate

One of the shared pathways was GPI-anchored protein biosynthesis (Fig. 4e), and one of the top hits among the shared ERAD substrates was PIGK (Fig. 6a). PIGK is the catalytic subunit of the ER-resident GPI–transamidase complex, which catalyzes the generation of nascent GPI-anchored proteins[42,67] (Supplementary Fig. 5a). PIGK contains a single-pass transmembrane domain and forms an intermolecular disulfide bond with PIGT (Supplementary Fig. 5a), with no predicted N-glycosylation site. PIGK was ubiquitously expressed in mouse organs and tissues (Supplementary Fig. 5b). PIGK protein level was elevated by 30–60% in ERAD-deficient HEK293T cells compared to that of WT cells (Fig. 6b), independent of transcriptional regulation (Supplementary Fig. 5c). This was confirmed using immunofluorescent staining (Supplementary Fig. 5d). Similarly, PIGK protein level was higher in *Sel1L*[Ucp1Cre] BAT by over 3 folds than that of WT BAT, without changes in mRNA levels (Fig. 6b and Supplementary Fig. 5c). Previous observations that many patients carrying pathogenic PIGK variants present a neuro-developmental syndrome[47] prompted us to further examine whether PIGK protein level is regulated by SEL1L–HRD1 ERAD in brains.

Using a tamoxifen-inducible *Sel1L*-deficient mouse model, we showed that endogenous PIGK protein levels were elevated by ~3 and 2 folds in the cortex and cerebellum, respectively, following acute SEL1L deletion (Fig. 6c), in keeping with the findings in HEK293T and BAT.

Mechanistically, endogenous PIGK and SEL1L physically interacted with each other in WT HEK293T cells, which was further enhanced when *HRD1* was ablated (Fig. 6d, e). In denaturing IP of PIGK, PIGK was found to be polyubiquitinated in an HRD1-dependent manner in cells treated with a proteasome inhibitor MG132 (Fig. 6f). Loss of either SEL1L or HRD1 stabilized endogenous PIGK protein in HEK293T cells (Fig. 6g). We concluded that nascent PIGK is a bona fide endogenous substrate of SEL1L–HRD1 ERAD.

We then asked how SEL1L–HRD1 ERAD recognizes PIGK. In the SEL1L-IP-MS proteomic screens, other subunits of the GPI–transamidase complex including PIGT, PIGS, PIGU, and GPAA1 were either not identified as high confidence hits or not immunoprecipitated by SEL1L (Supplementary Fig. 6a). Unlike PIGK, PIGT was unchanged after acute SEL1L deletion in the cortex and cerebellum of inducible *Sel1L*-deficient mouse model (Fig. 6c). In addition, in SEL1L-IP of HEK293T cells, PIGT–SEL1L interaction was not increased in the absence of HRD1, unlike that of PIGK-SEL1L (Fig. 6d). Lastly, unlike PIGK, PIGT, and three other subunits were not stabilized in HRD1 KO HEK293T cells (Fig. 6g and Supplementary Fig. 6b). Therefore, we concluded that SEL1L–HRD1 ERAD specifically targets PIGK protein for proteasomal degradation.

## SEL1L–HRD1 ERAD attenuates the biosynthesis of GPI-APs, at least in part, via PIGK

Intriguingly, PIGK–PIGT interaction was enhanced by 30% in SEL1L- or HRD1- deficient cells (Fig. 6e), pointing to a likely enhanced formation of the GPI–transamidase complex. We next asked whether SEL1L–HRD1 ERAD plays a role in the biogenesis of GPI-anchored proteins. Total GPI-anchored proteins at the cell surface probed with fluorescent aerolysin (FLAER) were elevated by 21–40% in ERAD-deficient cells compared to those of WT HEK293T cells (Fig. 7a). Indeed, surface abundance of a specific GPI-anchored protein CD59 was elevated by ~25% in ERAD-deficient cells (Fig. 7b). This increase was accompanied by an increase in intracellular CD59 following the cleavage of surface GPI-APs with a phosphatidylinositol (PI)-specific phospholipase C (PI-PLC) (Supplementary Fig. 7a, b). In line with PIGK being the catalytic subunit of GPI-transamidase complex[68], GPI-anchored proteins became undetectable at the cell surface of *PIGK KO* cells, while free GPI levels accumulated (Fig. 7a–c and Supplementary Fig. 7c). Moreover, deletion of PIGK in *SEL1L KO* cells abolished the surface accumulation of FLAER and CD59 proteins, while increased surface free GPI (Fig. 7d–f and Supplementary Fig. 7c), indicating that regulation of GPI-anchored protein biogenesis by SEL1L–HRD1 ERAD is PIGK-dependent. Restoring the expression of SEL1L or HRD1 in the corresponding ERAD-deficient cells reduced the surface levels of CD59, to a level similar to those in WT cells (Fig. 7g, h).

The effect of ERAD on GPI-APs was not related to ER stress as treatment with ER stress inducer thapsigargin (Tg) triggered strong ER stress but failed to increase surface CD59 level (Supplementary Fig. 7d, e) and as SEL1L- and HRD1-deficiency only caused subtle, if any, ER stress in HEK293T cells (Supplementary Fig. 7e). Hence, we concluded that SEL1L−HRD1 ERAD attenuates the biosynthesis of GPI-anchored proteins, at least in part, via PIGK.

**PIGK disease mutants are SEL1L−HRD1 ERAD substrates**

Lastly, we explored the clinical relevance of our findings by asking whether at least a subset of PIGK disease mutants in patients with neurodevelopmental syndrome or severe infantile encephalopathy are SEL1L−HRD1 ERAD substrates. We randomly picked five disease-causing PIGK variants in the N-terminal luminal domain of the protein (Fig. 8a). Notably, based on AlphaFold2 predicted PIGK structure,

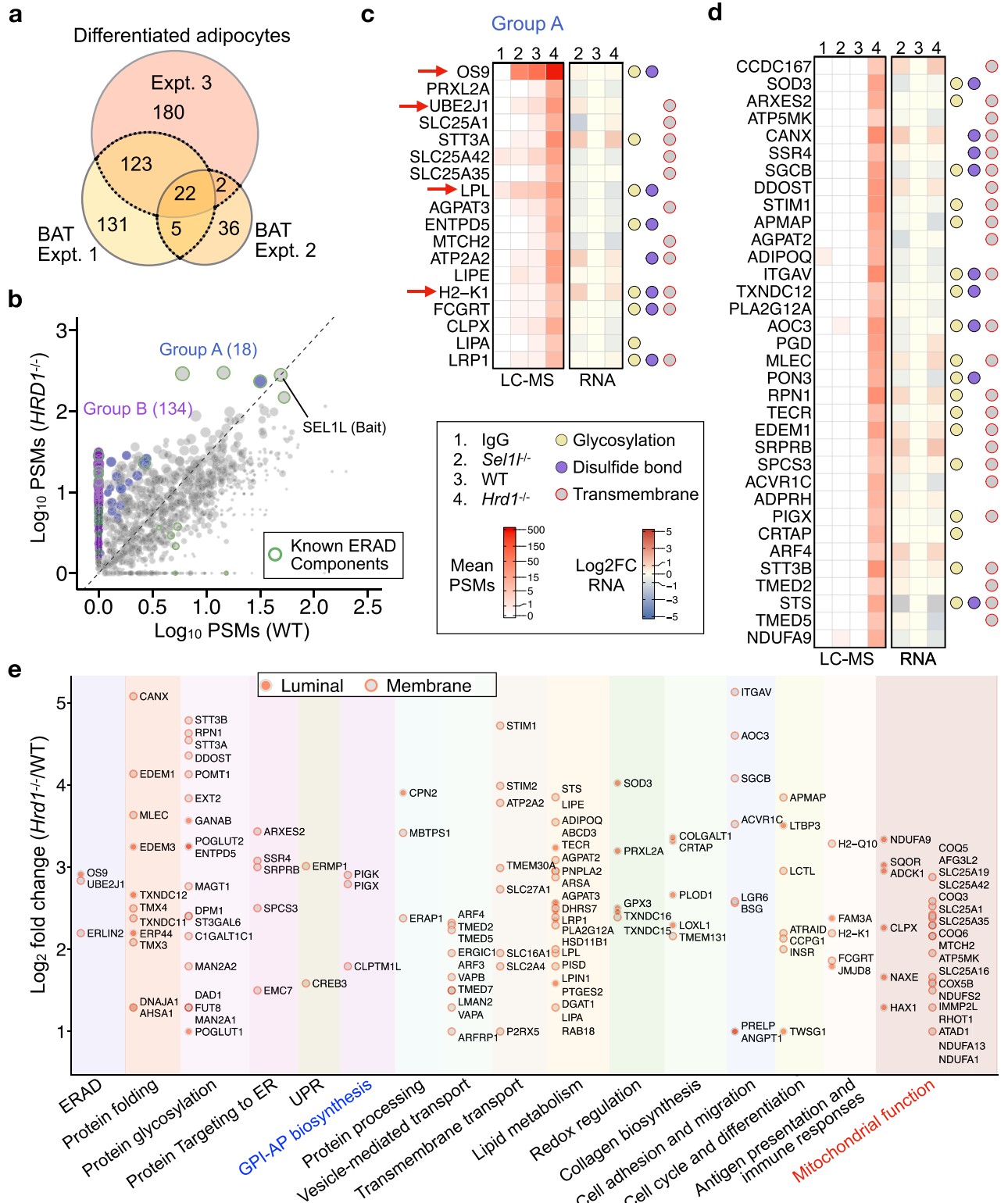

**Fig. 3 | Proteomic screening for SEL1L–HRD1 ERAD substrates in adipocytes.**
**a** Venn diagram showing the overlaps of substrate candidates identified from three independent experiments including two experiments from brown adipose tissues (BAT, $n = 4$ mice in Expt. 1 and 1 mouse in Expt. 2) and one experiment from differentiated white adipocytes. **b** Scatter diagram showing peptide-spectrum matches (PSMs) in WT and *Hrd1 KO* samples for each non-redundant protein hit from SEL1L-IP-MS in brown adipocytes and brown adipose tissue (BAT). The slope of the dashed black line denotes the cutoff value calculated from PSMs of the bait (SEL1L) in *Hrd1 KO* compared to WT samples used for substrate selection. ERAD substrate candidates detectable or non-detectable in WT samples are classified as Group A (in blue) and B (in purple). Dot size is proportional to the protein's mean

score from three experiments. Known ERAD pathway component proteins are highlighted with green circles. **c** and **d** Heatmaps showing mean PSMs from IgG, *Sel1L KO*, WT, and *Hrd1 KO* SEL1L-IP-MS samples and RNA log2 fold change (FC) in *Sel1L KO* and *Hrd1 KO* relative to WT for the top ERAD substrate candidates in Group A (**c**) and Group B (**d**). Dots on the right indicate the presence of protein N-glyco-sylation, disulfide bonds, and transmembrane domains. **e** Functional categorization and mean fold change in *Hrd1 KO* over WT samples for each ERAD substrate candidate. Luminal or membrane topology is annotated as filled or open red circles, respectively, for both ER and mitochondrial proteins. GPI-AP glycosylpho-sphatidylinositol-anchored protein.

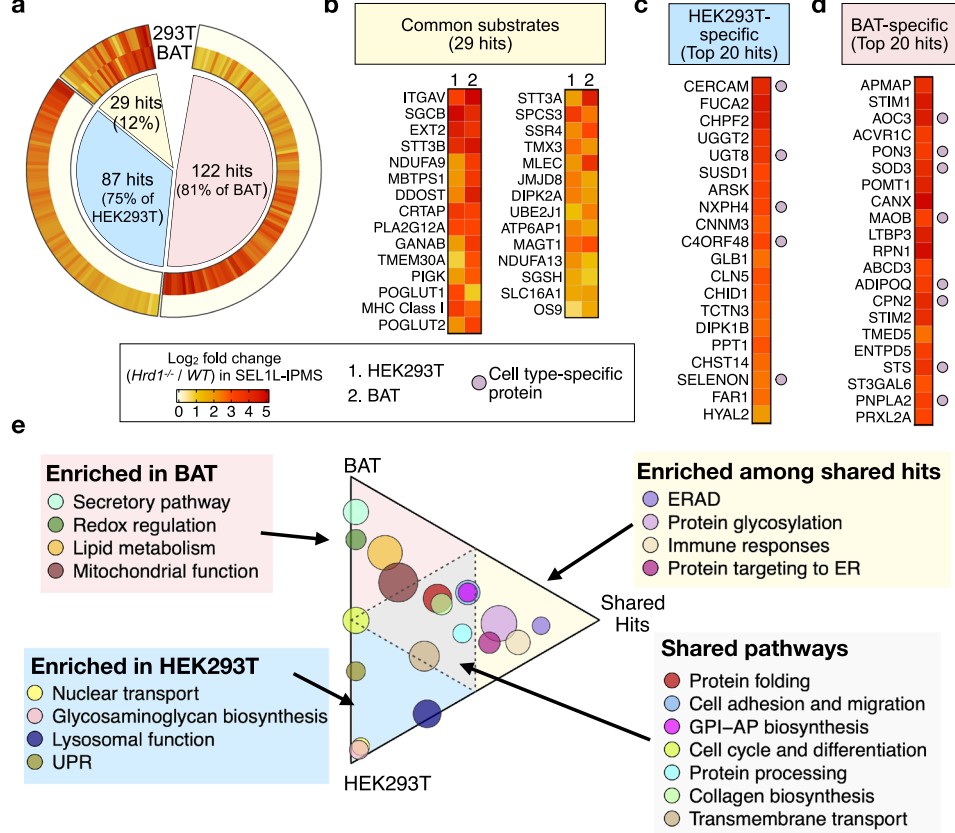

**Fig. 4 | Shared and cell type-specific putative SEL1L–HRD1 ERAD substrates.**
**a** Circular heatmap showing the overlaps of ERAD substrate candidates between HEK293T and brown adipose tissue (BAT). Counts and percentages for each group are indicated in a pie chart. Human HLA and mouse H2 proteins were grouped as MHC class I and counted once. The color scale indicates the mean fold change of PSMs in the *Hrd1 KO* samples over the WT samples on a logarithmic scale ($log_2$). **b–d** Heatmaps of log2 fold change (*Hrd1 KO/WT*) of PSMs in SEL1L-IP-MS for the 29 shared hits (**b**), the top 20 cell type-specific substrate candidates in HEK293T (**c**)

and BAT (**d**). Proteins expressed in a cell type-specific manner defined based on their transcript abundance from RNA-seq data are highlighted with a circle to the right of the heat maps. **e** Ternary plot showing the pathways enriched in the shared or cell type-specific substrate groups. The count of candidate substrate proteins in each group was normalized to the total number in the corresponding group. The cutoff for enriched and shared pathways was indicated as dashed lines in the ternary plot. Circle sizes are proportional to the number of hits affiliated with each pathway.

three mutants (L86P, A87V and D88N) are localized at the interface with N-terminal domain of PIGT[69], close to the intermolecular disulfide bond between PIGK and PIGT, hence may impair the assembly of the complex. On the other hand, Y160S and D204H are located at the catalytic site of PIGK[69], and therefore may affect the overall folding of the protein as well as its activity (Fig. 8b). All five variants were accumulated in *HRD1 KO* cells compared to WT cells (Fig. 8c) and degraded by SEL1L–HRD1 ERAD, as ablation of HRD1 significantly stabilized the mutant protein (Fig. 8d and Supplementary Fig. 8a). Moreover, PIGK variants formed high molecular weight (HMW) protein aggregates in transfected *HRD1 KO* cells, to a much more extent, than those in transfected WT cells (Supplementary Fig. 8b). These HMW complexes were formed via aberrant intermolecular disulfide bonds as they were

sensitive to the reducing agent β-mercaptoethanol (Supplementary Fig. 8b). These findings suggested that SEL1L–HRD1 ERAD degrades misfolded PIGK disease variants and limits the pathogenicity of the disease variants by preventing the formation of HMW aggregates.

## Discussion

SEL1L–HRD1 ERAD is a key quality-control mechanism for the clearance of misfolded ER proteins, thereby maintaining ER homeostasis. Although much advance has been made recently in elucidating its biochemical structure in yeast[4] and physiological importance in mammals[2,3,35], our current understanding of how SEL1L–HRD1 ERAD achieves such a prominent role under physiological conditions remains limited, largely due to our limited understanding of the nature

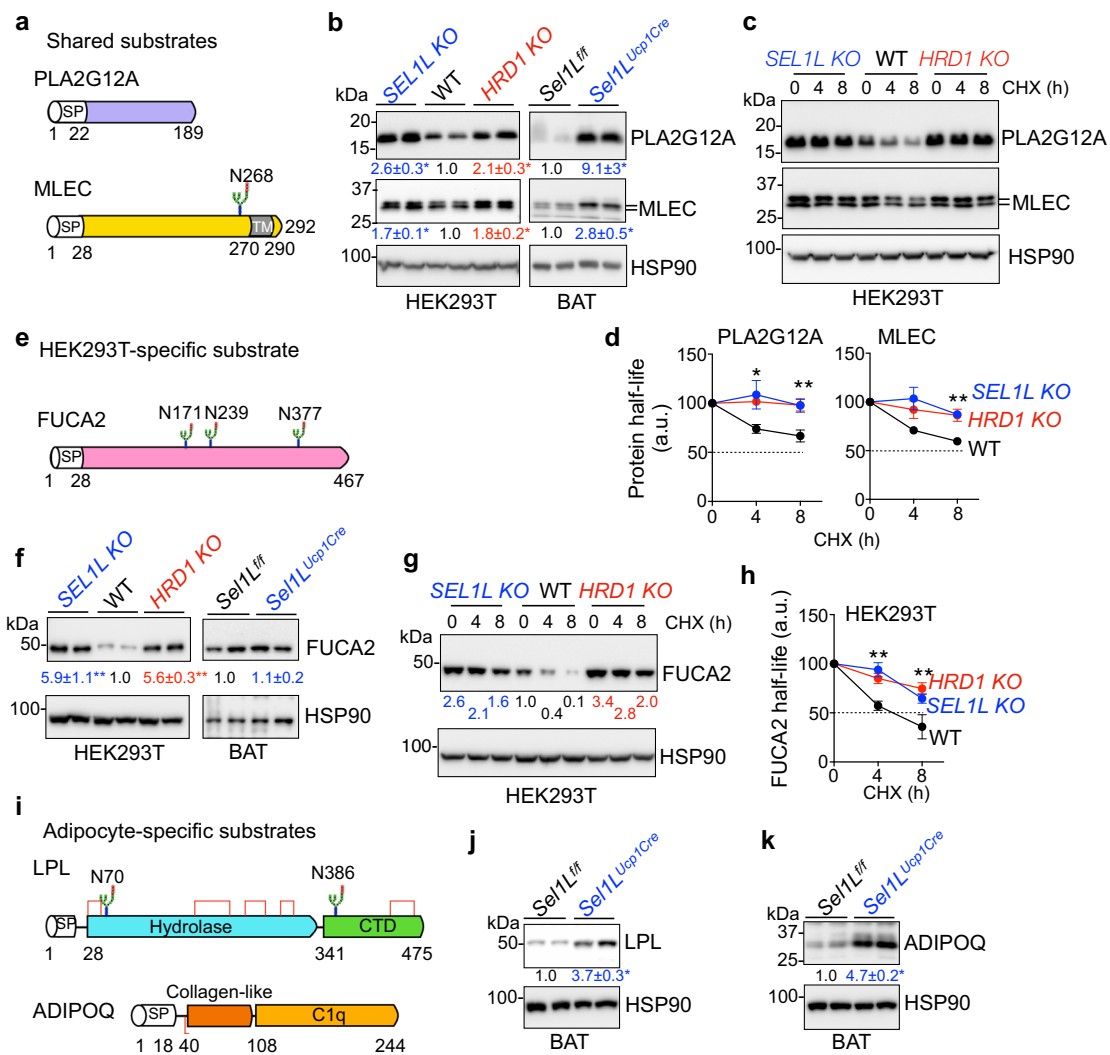

**Fig. 5 | Validation of a subset of shared or cell type-specific ERAD substrates.**
**a** Domain diagram for human PLA2G12A and MLEC. "SP", Signal Peptide; "TM",
Transmembrane. N-glycosylation site is highlighted by the asparagine residue
position. **b** Immunoblot analysis showing elevated protein abundance of PLA2G12A
and MLEC in ERAD-deficient HEK293T cells and brown adipose tissue relative to WT
samples with quantitation shown below the blot ($n = 4$–6 samples from 2 inde-
pendent experiments). **c** Immunoblot analysis showing ERAD-dependent degra-
dation of PLA2G12A and MLEC in HEK293T cells following cycloheximide (CHX)
treatment for the indicated durations, with quantitation from three independent
experiments shown in (**d**). a.u. arbitrary units. **e** Domain diagram for human FUCA2
with three N-glycosylation sites. **f** Immunoblot analysis in HEK293T cells and BAT

with quantitation shown below the blots ($n = 4$ samples from two independent
experiments). **g** Immunoblot analysis in HEK293T cells following CHX treatment for
the indicated durations, with quantitation shown in **h** ($n = 4$ independent repeats
for WT and *HRD1 KO*, 3 for *SEL1L KO*). a.u., arbitrary units. **i** Domain diagram for
human LPL and ADIPOQ with positions of N-glycosylation sites and disulfide bonds
shown. CTD C-terminal domain. **j** and **k** Immunoblot analysis showing elevated
protein abundance of LPL (**j**) and ADIPOQ (**k**) in BAT ($n = 3$-4 mice each). Values
represent mean ± SEM. *$P < 0.05$, **$P < 0.01$ using two-tailed one-way ANOVA fol-
lowed by Dunnett's multiple comparisons test (for HEK293T) and two-tailed Stu-
dent's *t*-test (for BAT). Source data are provided as a Source Data file.

of endogenous ERAD substrates in vivo. Here we report the develop-
ment of a robust label-free affinity purification MS-based method for
high-throughput screening of endogenous SEL1L−HRD1 ERAD sub-
strates both in vitro and in vivo (Fig. 8e). Distinct from previously
published studies, our design for proteomic screen of ERAD substrates
does not require any drug treatment (such as chemical inhibitors of
certain ERAD components−which is often fraught with off-target
effects) or isotope labeling for quantitation, making it compatible with
much broader sample types, especially tissues. By immunoprecipitat-
ing SEL1L in *HRD1 KO* cells or tissues, endogenous substrates in asso-
ciation with SEL1L are enriched in the IP samples (Fig. 8e). By
leveraging this intrinsic property of SEL1L−HRD1 ERAD, we achieved
high sensitivity in identifying endogenous substrates in the proteome
both in vitro and in vivo−over 100 potential high confidence sub-
strates of SEL1L−HRD1 ERAD in each cell type, implicating SEL1L−HRD1

ERAD in a wide array of cellular processes in different cell types. These
findings are in line with recent studies showing the profound sig-
nificance of SEL1L−HRD1 ERAD in a number of physiological processes
in various cell types[2,3,35].

A significant portion of the cell type-specific ERAD substrates
could likely be explained by their cell type-specific expression, such
as adipocyte-specific expression of LPL or adiponectin. However, we
also observed some substrates, such as FUCA2 expressed in both
HEK293T and BAT, which were only degraded by ERAD in HEK293T.
We speculate that differences in biosynthetic rates, expression of
specific chaperones, and/or post-translational modifications may
contribute to the cell type-specific substrate selection. In addition,
the presence of other degradative pathways, such as ER-phagy, may
compensate for the loss of SEL1L−HRD1 ERAD in a cell type-specific
manner[31].

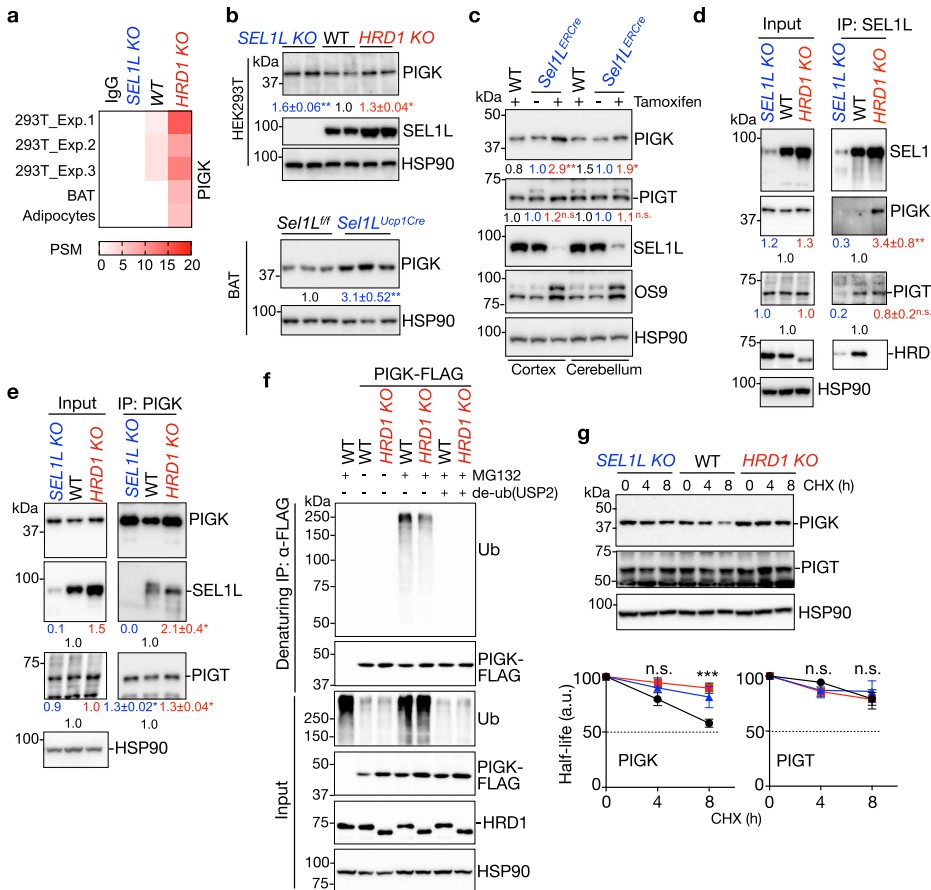

**Fig. 6 | PIGK is an endogenous SEL1L−HRD1 ERAD substrate. a** Heatmap showing the PSMs of PIGK from SEL1L-IP-MS experiments in HEK293T cells, BAT, and differentiated brown adipocytes. **b** Representative immunoblot analyses of endogenous PIGK in HEK293T cells (upper) and BAT (lower), with quantitation shown below the blots (*n* = 4−5/9 independent repeats for HEK293T/BAT). **c** Immunoblot analyses of endogenous PIGK and PIGT in the cerebral cortex and cerebellum of WT and *SellL*[ERCre] mice with or without 100 mg/kg tamoxifen injection (*n* = 4−5 from 3 independent repeats). **d** Representative Western blot analysis following IP of endogenous SEL1L in HEK293T cells with quantitation shown below the blots (*n* = 5 independent repeats). **e** Representative Western blot analysis following IP of endogenous PIGK in HEK293T cells showing the interactions between PIGK and SEL1L or PIGT (*n* = 4 independent repeats). **f** Immunoblot analyses of poly-ubiquitination following denaturing IP of PIGK-FLAG in transfected WT and *HRD1 KO* HEK293T cells treated with or without MG132 treatment for 2 h (*n* = 2 independent repeats). **g** Immunoblot analyses of endogenous PIGK or PIGT protein in HEK293T cells treated with 50 µg/ml cycloheximide (CHX) for the indicated times with quantitation shown below (*n* = 5/3 independent repeats for PIGK/PIGT). a.u. arbitrary units. Values represent mean ± SEM in **b**−**e** and **g**, mean in **c**. n.s. not significant, *$P$ < 0.05, **$P$ < 0.01, ***$P$ < 0.001 using two-tailed one-way ANOVA followed by Dunnett's multiple comparisons tests (for HEK293T, cortex and cerebellum) and two-tailed Student's *t*-test (for BAT). Source data are provided as a Source Data file.

In many of the cell types with impaired SEL1L or HRD1 function, the ER becomes dilated[5,30], presumably to adapt to the accumulation of misfolded proteins. Given the plausible activation of several compensatory mechanism(s), it is highly likely that substrates identified by SEL1L-IP-MS may be underestimated. In addition, a subset of substrate candidates may also be subjected to transcriptional regulation as a result of mild UPR observed in the ERAD-deficient cells, leading to the overestimation of the potential substrates. This indirect response to ERAD deficiency, however, is not mutually exclusive with the possibility that they are also targeted by SEL1L−HRD1 ERAD for degradation. Hence, vigorous validation for each candidate is needed.

We showed that SEL1L−HRD1 ERAD targets PIGK for proteasomal degradation, thereby negatively regulating the abundance of functional GPI-transamidase complex and hence the biogenesis of GPI-anchored proteins in the ER. Intriguingly, this effect of SEL1L−HRD1 ERAD on the activity of the GPI-transamidase complex seems specifically mediated through PIGK, not the other subunits (Fig. 8e). Providing further clinical relevance of our findings, SEL1L−HRD1 ERAD also degrades PIGK disease variants thereby preventing pathogenic aggregation. Given the profound importance of GPI-anchored proteins in health and disease[47], these findings may not only provide novel

insight into the pathological significance of SEL1L−HRD1 ERAD in this process but also help identify new therapeutic targets and strategies in the treatment of patients with PIGK deficiency[47,48].

The finding that SEL1L−HRD1 ERAD regulates the biogenesis of GPI-anchored proteins is surprising, as it has long been assumed that ER quality control seems limited for GPI-anchored proteins. A previous study in yeast has shown that following the addition of the GPI tag, most proteins, even those known to be misfolded, can evade the ER-quality control systems to efficiently exit the ER[70,71]. While our data did not explore whether and how many GPI-anchored proteins are direct ERAD substrates, our data showed that overall biogenesis of the GPI-anchored proteins in the ER is likely regulated by SEL1L−HRD1 ERAD. Moreover, in contrast to our findings, a recent study showed that PIGK is not an HRD1 substrate in WT HEK293T cells and only becomes an ERAD substrate when PIGT is absent[72]. The basis for the discrepancy is currently unclear.

In addition to candidate proteins that are ER-resident, our screen also identified a handful of candidates in both cell types that are mitochondrion-localized. It remains unclear how they became associated with SEL1L. Speculatively, they may be either mis-targeted to the ER[73,74] or transiting to the ER prior to reaching mitochondria[75] in their

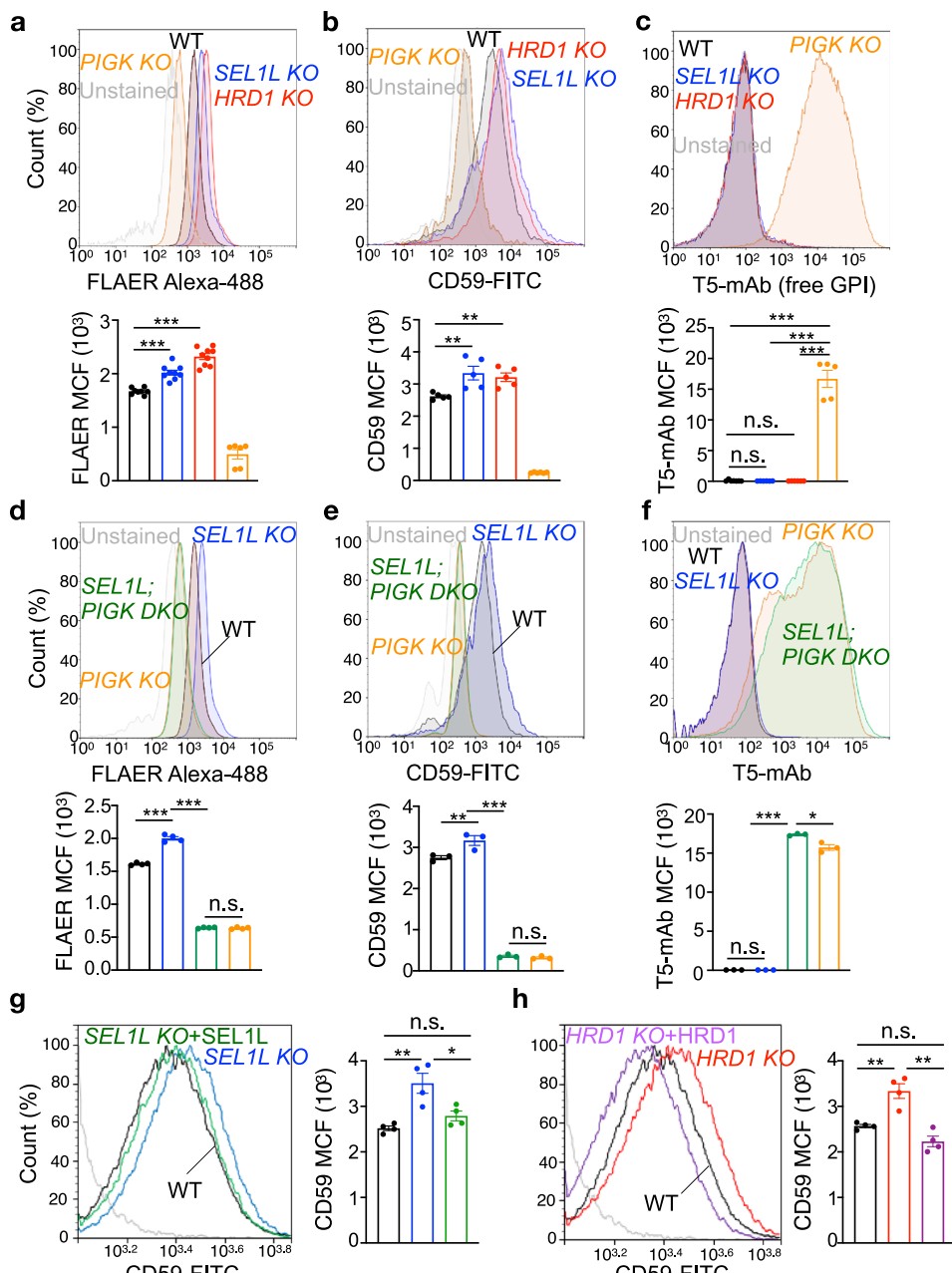

**Fig. 7 | SEL1L−HRD1 ERAD attenuates the biosynthesis of GPI-anchored proteins, at least in part, via PIGK. a–c** Flow cytometry analysis of total GPI-APs (FLAER), CD59 protein or free GPI (T5-mAb) in WT, *SEL1L KO*, *HRD1 KO,* and *PIGK KO* HEK293T cells, with quantitation shown below (*n* = 8 independent samples for WT, 9 for *SEL1L KO* and *HRD1 KO*, 6 for *PIGK KO* in **a**; *n* = 5 independent samples for WT, *SEL1L KO* and *HRD1 KO*, 6 for *PIGK KO* in **b**; *n* = 7 independent samples for WT, 6 for *SEL1L KO* and *HRD1 KO*, 5 for *PIGK KO* in **c**). **d–f** Flow cytometry analysis of total GPI-APs, CD59 or free GPI in WT, *PIGK KO, SEL1L KO, SEL1L; PIGK DKO* HEK293T cells,

with quantitation shown below (*n* = 4 independent samples in **d**, 3 in **e** and **f**). **g** and **h** Flow cytometry analysis of CD59 surface expression in *SEL1L KO* and *HRD1 KO* HEK293T cells transfected with SEL1L and HRD1, respectively, with quantitation shown on the right (*n* = 4 from two independent repeats). Values represent mean ± SEM. n.s., not significant, \**P* < 0.05, \*\**P* < 0.01 and \*\*\**P* < 0.001 by two-tailed one-way ANOVA followed by Dunnett's multiple comparisons test. Source data are provided as a Source Data file.

life cycle. In addition, given that mitochondria and the ER form physical tethers[76] and ERAD regulates mitochondrial fusion and fission dynamics[34], it is tempting to speculate that SEL1L−HRD1 ERAD may also selectively target proteins present in other organelles, such as mitochondrion, for proteasomal degradation. Taken together, while we acknowledge that a caveat of our proteomic screening is its limited ability to identify HRD1 cytosolic substrates, further studies are required to delineate the importance of SEL1L−HRD1 ERAD in the quality control of proteins in other organelles and its underlying mechanism(s).

## Methods

### Animals and animal experiments

The mice used in this study were age- and gender-matched littermates on the C57BL/6 J background. *Sel1L^flox/flox^*, *Sel1L^Ucp1Cre^* and *Sel1L^ERCre^* mice were described previously[5,33,34]. All animal procedures were approved by and done in accordance with the Institutional Animal Care & Use Committee (IACUC) at the University of Michigan Medical School (PRO00010658). The mice were housed in a pathogen-free animal facility at 22 °C with 40–60% humidity on a 12-h light/dark cycle. Brown adipocyte tissues were collected from 12 to 13-week-old mice at room

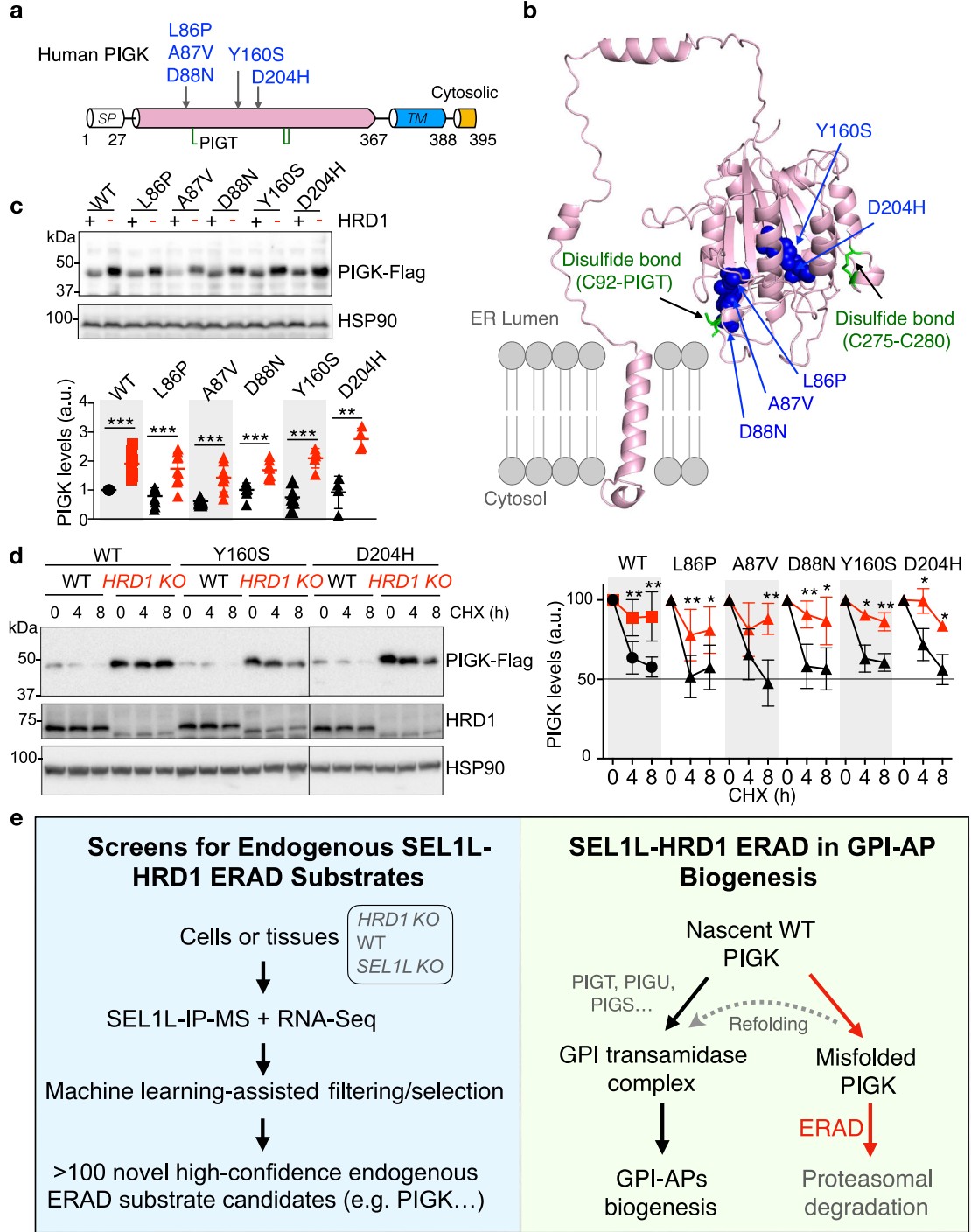

**Fig. 8 | A subset of PIGK disease mutants are SEL1L–HRD1 ERAD substrates.**
**a** Schematic illustration of human PIGK. "SP", Signal Peptide; "TM", Transmembrane. Green lines represent disulfide bonds. Five randomly selected point mutations associated with human disease are highlighted. **b** Structural modeling of human PIGK by AlphaFold2 showing the location of five pathogenic mutations highlighted in blue and 3 cysteine residues involved in disulfide bond formation in green. **c** Immunoblot analyses of WT and mutant PIGK in transfected WT and *HRD1 KO* HEK293T cells (n = 10 independent repeats for WT-PIGK, 9 for L86P, 8 for A87V and D88N, 5 for Y160S, 4 for D204H), with quantitation shown below. a.u. arbitrary units. **d** Representative immunoblot analyses of WT and mutant PIGK in transfected WT and *HRD1 KO* HEK293T cells treated with 50 µg/ml cycloheximide (CHX) for the indicated durations (n = 8 independent repeats for WT-PIGK, 6 for L86P, 5 for A87V and D88N, 3 for Y160S and D204H), with quantitation shown below. a.u. arbitrary units. **e** Graphic abstract of the paper: a label-free strategy to identify high confidence endogenous SEL1L–HRD1 ERAD substrates in two different cell types by combining SEL1L IP-MS, RNA-seq, and machine learning-assisted selection. One of the top shared candidates is PIGK, a catalytic subunit of the GPI transamidase complex. Misfolded PIGK protein is degraded by SEL1L–HRD1 ERAD, thereby limiting the abundance of functional GPI-transamidase complex and the production of GPI-APs. Values represent mean ± SEM. *P < 0.05, **P < 0.01 and ***P < 0.001 by two-tailed Student's *t* test. Source data are provided as a Source Data file.

temperature. For tamoxifen treatment, *Sel1L^flox/flox* and *Sel1L^ERCre* mice were intraperitoneally injected with 100 mg/kg tamoxifen (Sigma T2859-1G) for 5 consecutive days. Body weights were monitored daily. The mice were sacrificed on day 14 after the last injection.

## Cell culture and generation of knockout cell lines

HEK293T cells were originally obtained from ATCC. WT pre-adipocytes were isolated from brown adipose tissue of mice and immortalized using SV40 large T antigen as described previously[32,77]. Briefly, confluent pre-adipocytes were induced to differentiation in DMEM supplemented with 10% FBS and a cocktail containing 10 μg/mL insulin (Sigma), 0.5 mM IBMX (VWR), and 1 μM dexamethasone (Calbiochem), followed by differentiation with 10 μg/mL insulin (Sigma). HEK293T cells and adipocytes were cultured in Dulbecco's modified Eagle medium (Invitrogen 11995081) supplemented with 10% fetal bovine serum (FBS) (Fisher brand FB12999102) and 1% penicillin–streptomycin at 37 °C with a 5% $CO_2$ atmosphere. *SEL1L*-, *HRD1*- or *PIGK*-deficient HEK293T cells, and *Sel1L*- or *Hrd1*-deficient brown adipocytes were generated with the CRISPR/Cas9 system as previously described[34]. Cells transfected with empty plasmids without sgRNA were used as wild-type control. The sequences for sgRNA oligonucleotides are human *SEL1L* 5′-GGCTGAACAGGGCTATGAAG-3′, human *HRD1* 5′-GGACAAAGGCCTGGATGTAC-3′, human *PIGK* 5′-GGCTCTAGCTAGTAGTCAAG-3′. PIGK-SEL1L-DKO cells were generated by the CRISPR/Cas9 system with two different sgRNA oligonucleotides. The sgRNA oligonucleotides for generating *Sel1L*- and *Hrd1*-deficient adipocytes are as follows: mouse *Sel1L* 5′-GAGCATAGGACACTCTCTCC-3′, mouse *Hrd1* 5′-GTACGCCATTCTGATGACCA-3′.

## Immunoprecipitation (IP)

Mouse tissues or cells were harvested, snap-frozen in liquid nitrogen and sonicated in lysis buffer [150 mM NaCl, 0.2% Nonidet P-40 (NP40), 0.1% Triton X-100, 25 mM Tris–HCl pH 7.5] with protease inhibitors (Sigma-Aldrich, P8340), protein phosphatase inhibitors (Sigma-Aldrich, P5726) and 10 mM N-ethylmaleimide. Lysates were incubated on ice for 30 min and centrifuged at 16,000 × g for 10 min. Supernatants were collected and analyzed for protein concentration using the Bio-Rad Protein Assay Dye (Bio-Rad, 5000006). A total of ~5 mg protein lysates were incubated with 10 μl anti-SEL1L (home-made)[34], anti-PIGK (Abcam, ab201693) or normal rabbit IgG (Cell Signaling Technology, #2729) overnight at 4 °C with gentle rocking. On the following day, lysates were incubated with protein A agarose (Thermo Fisher Scientific, 5918014) for 6 h, washed three times with lysis buffer, and eluted in the 5× SDS sample buffer (250 mM Tris–HCl pH 6.8, 10% sodium dodecyl sulfate, 0.05% Bromophenol blue, 50% glycerol, and 1.44 M β-mercaptoethanol) at 95 °C for 5 min followed by SDS–PAGE and immunoblot.

## IP-based mass spectrometry

HEK293T cells or differentiated white adipocytes were collected and pooled from 4 × 10 cm culture dishes. IP in BAT was performed using frozen BAT from 12-week-old mice (n = 4 in Expt. 1; and n = 1 in Expt. 2). Cells or tissues were lysed with cold IP lysis buffer [150 mM NaCl, 0.2% Nonidet P-40 (NP40), 0.1% Triton X-100, 25 mM Tris–HCl pH 7.5] with protease inhibitors (Sigma-Aldrich, P8340), protein phosphatase inhibitors (Sigma-Aldrich, P5726). 10 mg protein lysates for each sample were incubated with anti-SEL1L or IgG overnight at 4 °C and then incubated with Protein A agarose beads for 4 h at 4 °C. After washing the beads with cold IP buffer three times, samples were submitted to the Proteomics Resource Facility at the University of Michigan Medical School on a fee-for-service basis. Briefly, the beads were resuspended in 50 μl of 0.1 M ammonium bicarbonate buffer (pH-8). Disulfide bonds in proteins were reduced by adding 50 μl of 10 mM DTT and incubating at 45 °C for 30 min. Samples were cooled to room temperature and alkylation of cysteines was achieved by incubating

with 65 mM 2-chloroacetamide, under darkness, for 30 min at room temperature. An overnight digestion with 1 μg sequencing-grade modified trypsin was carried out at 37 °C with constant shaking in a Thermomixer. Digestion was stopped by acidification and peptides were desalted using SepPak C18 cartridges using the manufacturer's protocol (Waters). Samples were completely dried using vacufuge. Resulting peptides were dissolved in 0.1% formic acid/2% acetonitrile solution and were resolved on a nano-capillary reverse phase column (Acclaim PepMap C18, 2 micron, 50 cm, ThermoScientific) using a 0.1% formic acid/2% acetonitrile (Buffer A) and 0.1% formic acid/95% acetonitrile (Buffer B) gradient at 300 nl/min over a period of 180 min (2–25% buffer B in 110 min, 25–40% in 20 min, 40–90% in 5 min followed by holding at 90% buffer B for 10 min and requilibration with Buffer A for 30 min). Eluent was directly introduced into Q exactive HF mass spectrometer (Thermo Scientific, San Jose, CA) using an Easy-Spray source. MS1 scans were acquired at 60 K resolution (AGC target = $3 \times 10^6$; max IT = 50 ms). Data-dependent collision-induced dissociation MS/MS spectra were acquired using Top speed method (3 s) following each MS1 scan (NCE ~ 28%; 15K resolution; AGC target $1 \times 10^5$; max IT 45 ms).

Proteins were identified by searching the MS/MS data against UniProt entries using Proteome Discoverer (v2.4, Thermo Scientific). Search parameters included: MS1 mass tolerance of 10 ppm and fragment tolerance of 0.2 Da, two missed cleavages were allowed, carbamidomethylation of cysteine was considered fixed modification and oxidation of methionine, and deamidation of asparagine and glutamine were considered as potential modifications. False discovery rate (FDR) was determined using Percolator and proteins/peptides with an FDR of ≤1% were retained for further analysis.

## Denaturing IP for ubiquitination assay

HEK293T cells were transfected with PIGK-FLAG plasmids for 16 h and then treated with 10 μM MG132 for the last 2 h. Cells were snap-frozen in liquid nitrogen and whole cell lysate was prepared in the NP-40 lysis buffer [50 mM Tris–HCl at pH 7.5, 150 mM NaCl, 1% NP-40, 1 mM EDTA], followed by centrifugation at 16,000 × g for 10 min. Supernatant was treated with 200 nM USP2 at 37 °C for 1 h and then denatured with 1% SDS and 5 mM DTT at 95 °C for 10 min. Subsequently, supernatants were diluted 1:10 with NP-40 lysis buffer and incubated with 15 μl anti-FLAG agarose (Thermo Fisher, 26182) overnight at 4 °C with gentle rocking. Agarose beads were washed three times with NP-40 lysis buffer and eluted in the SDS sample buffer at 95 °C for 5 min, followed by SDS–PAGE and immunoblot.

## Western blot

Mouse tissues or cells were harvested and processed as described above. 20–50 μg of protein were denatured at 95 °C for 5 min in 1× SDS sample buffer. For PIGK experiments, protein lysates were denatured at 37 °C for 5 min. SDS–PAGE was performed and followed by electrophoretic transfer to the PVDF membrane (EMD Millipore, IPVH00010). The blots were incubated with primary antibodies in 2% BSA containing Tris-buffered saline with 0.1% tween-20 (TBST) overnight at 4 °C. On the following day, membranes were washed with TBST for three times and incubated with secondary antibodies for 1 h at room temperature. After washing three times with TBST, membranes were incubated with Clarity Western ECL Substrate (Bio-Rad, 1705061). The chemiluminescence detection system (Bio-Rad) was used to collect images and quantify band intensity. The primary antibodies were described as below: anti-HSP90 (Santa Cruz, sc-7947, 1:5000), anti-SEL1L (home-made, 1:10,000)[34], anti-HRD1 (Proteintech, 13473-1, 1:2000), anti-OS9 (Abcam, ab109510, 1:5000), anti-PLA2G12A (Proteintech,16009-1-AP, 1:2000), anti-MLEC (Proteintech, 26655-1-AP, 1:1000), anti-FUCA2 (Proteintech, 15157-1-AP, 1:1000), anti-LPL (Novus Biologicals, AF7197, 1:500), anti-PIGK (Abcam, ab201693, 1:2000), anti-Ubiquitin (Santa Cruz, sc-8017, 1:1000), anti-HA (Sigma-Aldrich,

H3663, 1:1000), anti-FLAG (Sigma-Aldrich, F1804, 1:1000), anti-PIGT (Proteintech, 16906-1-AP, 1:1000), anti-PIGU (Abclonal, A18538, 1:1000), anti-PIGS (Proteintech, 18334-1-AP, 1:1000), anti-GPAA1 (Santa Cruz Biotechnology, sc-373710, 1:500). Source data are provided as a Source Data file.

## Identification of SEL1L-interacting proteins in HEK293T and comparative analyses

SEL1L-interacting proteins were selected from the SEL1L-IP-MS data in WT and *SEL1L KO* HEK293T. ER-resident or membrane protein hits with PSMs in *SEL1L KO* negative control smaller than one-tenth of the PSMs in WT samples were considered as SEL1L-interacting proteins. To compare with previously reported SEL1L-interacting proteins, a reference list was compiled as a union set from the String database using SEL1L as a query to search for direct physical interactors in the human proteome with high confidence score (>0.7), and the proteins hits from previously published proteomics screening studies in HEK293T using overexpressed baits[8,41,57]. Specifically, 58 SEL1L-interactors in Fig. 1d were generated using the OS9-interacting proteins reported in Tyler et al.[41], HRD1-interacting proteins reported in Fenech et al.[57], and the SEL1L-centered ERAD network reported in Christianson et al.[8], which was reconstituted using prey and bait proteins that either directly interact with SEL1L or indirectly via only one intermediate interactor. The SEL1L-interacting proteins identified in this study and not in the overlap were considered as novel SEL1L-interacting proteins.

## ERAD substrate candidate selection

The peptide-spectrum match (PSM) counts data were processed using a data-driven machine learning method to identify high-confidence ERAD substrate candidates. Based on the published literature, we found out and labeled known ERAD substrates such as SIGMAR1, HLA, OS9, and so on as positive hits in our datasets. The hits that do not have any signal peptide, transmembrane domain, glycosylation, or disulfide bonds, or are not localized in ER, Golgi, lysosome, or membrane, were labeled as negative hits. Logistic regression was performed and confidence score $y$ was calculated based on the following equation:

$$y = \frac{1}{1 + \exp\left[-(w_0 + w_1 \cdot x_1 + w_2 \cdot x_2)\right]} \quad (1)$$

$$x_1 = \frac{hko}{\max(wt, IgG)} \quad (2)$$

$$x_2 = \frac{hko}{\max(sko, IgG)} \quad (3)$$

Here, $y$ is the confidence score, while $w_1$ and $w_2$ are the weight factors for feature $x_1$ and feature $x_2$. *hko* represents the PSM value in the HRD1 knock-out cell or tissue, *sko* represents the PSM value in the SEL1L knock-out cell or tissue and IgG represents the PSM value in the IgG control sample. Coefficients of the model ($w_0$, $w_1$, and $w_2$) were optimized on all labeled hits by gradient descent to maximize the Matthews correlation coefficient (MCC):

$$MCC = \frac{TP \cdot TN - FP \cdot FN}{\sqrt{(TP+FP)(TP+FN)(TN+FP)(TN+FN)}} \quad (4)$$

Here TP, TN, FP, FN are true positive, true negative, false positive, and false negative, respectively. The value of MCC ranges from −1 to 1. Higher MCC indicates higher consistency between prediction and known positive/negative labels.

After getting the confidence score for each hit, the cutoff values for the confidence score were searched with the lowest score from hits identified only in the *HRD1 KO* samples with 2 PSMs.

After removing hits derived from keratin (KRT) and keratin-associated protein (KRTAP), each hit was subsequently filtered based on the following criteria: PSM ratio of *HRD1 KO* to WT greater than the smaller value of the *HRD1 KO* to WT ratio of SEL1L or OS9 in the same experiment; PSM of *SEL1L KO* must be smaller than WT unless PSM of *SEL1L KO* is no >1; PSM of *HRD1 KO* must be greater than PSM of *SEL1L KO*; PSM of IgG must be zero or no greater than one-tenth of PSM of *HRD1 KO*. Nucleus-localized proteins were excluded unless it contains any signal peptide, N-glycosylation, disulfide bonds, or transmembrane domains. The hits that passed the criteria in at least two independent experiments were considered ERAD substrate candidates.

## Cataloging and comparison of common vs. cell type-specific ERAD substrate candidates

To catalog common versus cell type-specific ERAD substrate candidates, all mouse gene symbols were first converted to corresponding human orthologs' symbols according to the Alliance of Genome Resources (https://www.informatics.jax.org/downloads/reports/HOM_MouseHumanSequence.rpt). The genes encoding the heavy chain of class I major histocompatibility (MHC-I) were grouped together as one hit for comparison purposes, as one-to-one orthologous relationship cannot be established between human HLA- and mouse H2- genes. ERAD substrate candidates present in both HEK293T and BAT are classified as common or shared, while ERAD substrate candidates present in only one cell type are considered as cell type-specific.

Pathway and subcellular location information for each hit were manually assigned based on the consensus of UniProt database records and literature. For pathway overrepresentation analysis, the numbers of hits associated with a certain pathway in the common, HEK293T-specific, and BAT-specific groups were first normalized to the total number of hits in each group. If the normalized number from one group represented over 50% of the sum of all three groups, then the pathway was considered overrepresented in that group; Otherwise, the pathway was considered as shared among the three groups. In the case of MHC-I, the larger of actual counts of HLA- or H2- hits were used as input for the common group.

## RNA-seq and quantitative PCR

Total RNA was extracted using the TRI Reagent (Molecular Research Center, TR118) and Phase Separation Reagent (Molecular Research Center, BP151). All the RNA-seq libraries were sequenced pair-ended for 151 cycles on the same flow cell using a NovaSeq 6000 instrument (Illumina). The resulting sequencing reads were mapped to the reference gene models GRCh38 (ENSEMBL) for HEK293T samples and to GRCm38 (ENSEMBL) for mouse brown adipose tissue samples using STAR aligner[78](version 2.7.8a) and converted to count estimates with RSEM[79](version 1.3.3). The downstream analysis for differentially expressed genes was conducted using DESeq2[80](version 1.36.0).

For performing quantitative PCR, the RNA samples were first reverse transcribed into cDNA with SuperScript III Reverse Transcriptase (Invitrogen, 18080-093) following the manufacturer's recommended protocol. Transcript abundance was measured with quantitative PCR using 2X Universal SYBR Green Fast qPCR Mix (ABclonal, RK21203) on a CFX Opus 384 Real-Time PCR System (Bio-Rad, 12011452). The ribosomal protein L32 gene was used as the housekeeping reference gene in the relative quantitation of transcript abundance following the $2^{-\Delta\Delta CT}$ approach.

Mouse *Rpl32* forward: 5′ GAGCAACAAGAAAACCAAGCA 3′, reverse: 5′ TGCACACAAGCCATCTACTCA 3′. Human *RPL32* forward: 5′ AGTTCCTGGTCCACAACGTC 3′, reverse: 5′ TTGGGGTTGGTGACTCTGAT 3′. Mouse *Pla2g12a* forward: 5′ TCCACAAGATAGACACGTACCTC 3′, reverse: 5′ GTTGTCTCACATGCCTGGAC 3′. Human *PLA2G12A* forward: 5′ CAGCATGTTCAGGCATGTGA 3′, reverse: 5′ CTGTCACTAGCTGTCGGCAT 3′. Mouse *Mlec* forward: 5′ CCTCGGACTATGGCATGAAAC 3′, reverse: 5′ ACTTCGTAGCCAAAGGTCTCTT 3′. Human *MLEC* forward: 5′

 

CTGGGGCAGTGGACATCCTA 3', reverse: 5' CCCCCTCCTTCCTCAG ACAG 3'. Mouse *Fuca2* forward: 5' CACTCCGGATGTGTGGTACA 3', reverse: 5' CCAATGGCCCAGCAGTTCTA 3'. Human *FUCA2* forward: 5' CTGGCGATCCCAGAATGACA 3', reverse: 5' GGCCCAGTAGTTTCACC TCT 3'. Mouse *Lpl* forward: 5' CGAGAGGATCCGAGTGAAAG 3', reverse: 5' TTTGTCCAGTGTCAGCCAGA 3'. Mouse *Adipoq* forward: 5' GGAAC TTGTGCAGGTTGGAT 3', reverse: 5' GCTTCTCCAGGCTCTCCTTT 3'. Mouse *Pigk* forward: 5' GCTGGACACATCGAGGATCA 3', reverse: 5' CGGGATGTGCACACCAAAAC 3'. Human *PIGK* forward: 5' TACTTGCC AAGGAGCATCCA 3', reverse: 5' CAGGATCAGGTTGATGCGAGA 3'.

## Immunofluorescence staining and confocal microscopy

The HEK293T cells cultured in 8-well Cell Culture Slides (MatTek, CCS-8) were fixed with 4% paraformaldehyde (pH 7.4) for 10 min at room temperature (RT) and permeabilized with 0.1% (v/v) Triton-X 100 (Thermo Fisher, BP151) in PBS for 15 min at RT. Samples were blocked using 5% (v/v) normal donkey serum in PBS for 1 h at RT and were incubated with the primary antibodies anti-PIGK (Abcam, ab201693, 1:500) and anti-KDEL (Novus Biologicals, NBP1-97469, 1:500) at 4 °C overnight. On the next day, slides were washed in TBST three times each for 10 min at RT, and incubated with secondary antibodies Alexa Fluor 488 AffiniPure Anti-Rabbit IgG (Jackson ImmunoResearch Laboratories, 711-545-152, 1:500) or Alexa Fluor 594 AffiniPure Anti-Mouse IgG (Jackson ImmunoResearch Laboratories, 115-585-044, 1:500) for 2 h at RT. Slides were washed three times for 10 min before mounting using ProLong Gold Antifade Reagent with DAPI (Invitrogen, P36931). Cells were imaged under the same setting using a Leica STELLARIS 8 FALCON confocal microscope.

## Flow cytometry

The HEK293T cells were harvested and fixed with 4% paraformaldehyde (PFA) and incubated with anti-human CD59 antibody conjugated with FITC (BioLegend 304706, 1:100) or FLAER (Alexa 488 proaerolysin variant, 1:100) on ice for 25 min. After two washes with PBS, the cells were analyzed with an Attune Flow Cytometer (Thermo Fisher Scientific). For free GPI test, cells were incubated with T5 mAb (BEI Resources, NR-50267, 1:100) on ice for 25 min and then washed twice followed by staining with Alexa Fluor 647-conjugated goat anti-mouse IgM secondary antibody (Invitrogen A-21238, 1:1000) prior to flow cytometry analysis. For phosphatidylinositol-specific phospholipase C (PI-PLC) treatment, cells were harvested and incubated with 5 U/mL PI-PLC in DMEM plus 0.5% BSA (made in PBS) for 1.5 h at 37 °C. Cells were washed twice with PBS prior to flow cytometry analysis.

## Protein structure analysis

PIGK structure was predicted using AlphaFold2 (https://alphafold.ebi.ac.uk/)[81], as certain regions of PIGK were not solved in the GPI−transamidase complex structure. All the protein structure images were rendered with PyMOL (version 2.3.2).

## Statistical analysis

All experiments were independently repeated two to five times. Plotted values and error bars represent the mean and standard error of the mean (SEM), respectively, unless otherwise noted. All datasets passed normality and equal variance tests. Statistical comparisons between two groups were conducted using two-tailed unpaired Student's $t$-test and for multiple groups were conducted using two-tailed one-way ANOVA followed by *post hoc* Dunnett's multiple comparisons test. All the statistical analyses were performed in GraphPad Prism version 9.0. $P = 0.05$ was considered as the threshold for statistical significance.

## Reporting summary

Further information on research design is available in the Nature Portfolio Reporting Summary linked to this article.

## Data availability

IP-MS datasets for mouse adipocytes and HEK293T cells are available via ProteomeXchange with identifiers PXD041803 and PXD041882, respectively. High-throughput RNA sequencing data have been deposited to the Gene Expression Omnibus (GEO) database under identifier GSE231583. The materials and reagents used are either commercially available or available upon request. All other data are available in the main text or in the Supplementary Information and Source Data Files. Source data are provided in the paper. Source data are provided in this paper.

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

## Acknowledgements

We thank Dr. Lydia P. Freddolino for technical assistance and members in the Arvan/Qi laboratories for insightful discussions; the Proteomics Resource Facility, Advanced Genomics Core and Microscopy and Image Analysis Core at the University of Michigan for assistance; and the BEI Resources for sharing T5 mAb. This work used the Services & Support (ACCESS) program of Advanced Cyberinfrastructure Coordination Ecosystem (National Science Foundation 2138259, 2138286, 2138307, 2137603, and 2138296) and was supported by 1R01DK128077, 1R01DK132068 (S.S.), 1R01DK120047, DK120330 and 1R35GM130292 (L.Q.). X.W. was/is supported in part by Pandemic Research Recovery Grant U078128 at the University of Michigan Medical School and the American Society of Nephrology Postdoctoral Fellowship. L.L.L. and Z.J.L. are supported in part by the National Ataxia Foundation Post- and Pre-doctoral Fellowship (NAF 918037 and 1036307). S.A.W. was supported by the American Heart Association Predoctoral Fellowship (828841) and Barbour Scholarship at the University of Michigan.

## Author contributions

X.W., Y.L., and L.L.L. collaboratively designed and performed most experiments; Y.L. and X.W. performed data analysis; L.L.L. performed IP-MS; X.W. performed most experiments related to PIGK; X.W. and C.Z. built the website for hosting the searchable databases; X.C., S.W., S.A.W., Z.J.L., and Y.Q. assisted with some experiments; X.W. performed the structural analysis; S.S. and L.Q. directed the study; X.W., Y.L., and L.Q. wrote the manuscript; all authors commented on and approved the manuscript.

## Competing interests

The authors declare no competing interests.
