## [Peer Review File · Nature Communications]

Proteomic screens of SEL1L-HRD1 ER-associated degradation substrates reveal its role in glycosylphosphatidylinositol-anchored protein biogenesisREVIEWER COMMENTS

Reviewer #1 (Remarks to the Author):

Authors developed a new proteomic screen for endogenous substrates of SEL1L-HRD1 ERAD useful for both cultured cells and tissues/organs. The developed method was applied to HEK293T cells and brown adipose tissue from mice, and was proven to be highly efficient in picking up candidates of SEL1L-HRD1 ERAD substrates. In fact, more than 100 candidates were obtained from both sources and ~20 % were common and the rests were cell-type specific. Five selected candidates from the common type were experimentally validated using SEL1L-KO and HRD1-KO HEK293T cells and/or Sel1L-defective mice, confirming usefulness of the developed method. The substrate candidates of known functions are involved in various cellular pathways including GPI-anchored protein biogenesis. Authors then focused on PIGK, the catalytic component of GPI transamidase complex that mediates attachment of GPI to precursor proteins, and demonstrated PIGK is a substrate of SEL1L-HRD1 ERAD. Based on a series of experiments, authors conclude that SEL1L-HRD1 ERAD suppresses biogenesis of GPI-anchored proteins by targeting PIGK for proteasomal degradation.

Application of the developed method to various tissues/organs should be useful to determine endogenous SEL1L-HRD1 ERAD substrates specific to respective tissues/organs.

Concerning the part of SEL1L-HRD1 ERAD effects on biogenesis of GPI-anchored proteins, conclusions are not fully supported by the provided results. I have following points to be addressed by authors.

Major points:

1. It is clearly shown in Fig 6 that PIGK protein levels increased under SEL1-HRD1 ERAD defective conditions in HEK293T cells, and mouse brown adipose tissue, cerebral cortex and cerebellum. In contrast, PIGT protein levels did not increase after SEL1L/Sel1L-KO or HRD1-KO. When PIGK was immunoprecipitated from SEL1L-KO, HRD1-KO and WT HEK293T cells, co-precipitated PIGT levels seem to be similar in three cells (Fig 6e). These results suggest that levels of GPI transamidase complex containing PIGK and PIGT did not increase under SEL1-HRD1 ERAD defective conditions and that PIGK free from GPI transamidase increased. It is likely that PIGK is synthesized in excess relative to PIGT in WT cells and excessive fraction of PIGK is degraded by SEL1-HRD1 ERAD.

2. Both cell surface levels of CD59, a GPI-AP, and staining levels by FLAER, a fluorescent probe for GPI moiety in some GPI-APs, increased in SEL1L-KO and HRD1-KO HEK293T cells. Increase was modest but was dependent upon SEL1L- and HRD1-KO because the increased CD59 levels turned to WT levels after rescue by corresponding cDNA (Fig 7). Because data in Fig 6 did not support an increase of GPI

transamidase, a possible reason for the increased cell surface levels of CD59 and FLAER staining might be related to stressed ER conditions in SEL1L-HRD1 ERAD defective cells. It is not surprising if trafficking/turnover of CD59 is modified in SEL1L-HRD1 ERAD defective cells, resulting in modest increase in the cell surface level.

Without showing increase of GPI transamidase complex or increased generation of GPI-anchored CD59 in the ER, the conclusion that SEL1L-HRD1 ERAD suppresses biosynthesis of GPI-APs via PIGK should be modified.

Other points:

1. It is unusual to use -/- for KO in HEK293T cells. Symbols +/+, +/- and -/- are used when cells/organisms are diploid where -/- means both alleles are dead. HEK293T cells are generally triploid. Only when SEL1L and HRD1 genes in HEK293T are exceptionally diploid and biallelic mutations are shown, -/- is appropriate. Authors showed complete loss of SEL1L and HRD1 proteins after KO, suggesting successful KO of all alleles. I suggest to use SEL1L-KO instead of SEL1L-/-.

2. Lines 104-105. This statement does not fit with data shown in Fig 6. If SEL1L-HRD1- ERAD is involved only in degradation of misfolded PIGK, how it plays a key role in GPI-AP biogenesis.

3. Fig 3e and related file.

CLPTM1 is listed as a factor in GPI-AP biogenesis. As far as I know, it has never been shown.

4. Line 271, "GPI-transamidase complex catalyzes the last step of biosynthesis of GPI anchored proteins" is not accurate. The enzyme generates nascent GPI-anchored proteins, whose GPI-anchor moiety is structurally immature and undergoes several maturation reactions in later steps.

5. Fig 6d. In IP: SEL1L part, SEL1L band in WT cell sample is very faint. PIGK is faint whereas PIGT is stronger than one in HRD1-KO. Please explain these unexpected profiles.

6. Fig 6f. When PIGK gets polyUb, PIGK should be seen as smear in the presence of MG132. Why only intact PIGK is shown?

7. Lines 295-302. Authors asked whether SEL1-HRD1 ERAD recognizes PIGK in a complex or as an individual protein. An answer to this question is not clearly stated in this paragraph.

8. M & M. Section “Immunoprecipitation. A lysis buffer contains 0.2% NP40 and 0.1% Triton-X100. These two detergents are similar. Is there any specific reason for mixing them?”

Reviewer #2 (Remarks to the Author):

In this work, the authors employed label-free MS-based proteomics to identify the substrates of Endoplasmic Reticulum-Associated Degradation (ERAD). In the SEL1L-HRD1 complex, which represents the most conserved branch of ERAD, SEL1L serves as a scaffolding protein for the ER-resident E3 ubiquitin ligase and retrotranslocon HRD1, and it recruits substrates. Using an antibody against SEL1L, they comprehensively identified the interactors of SEL1L. Combining a computational method for stringent filtering, more than 100 potential substrates were identified in both the human kidney cell line (HEK293T) and the mouse brown adipose tissue, respectively. They further validated several potential substrates, i.e., PLA2G12A, MLEC, FUCA2, ADIPOQ and LPL, as endogenous SEL1L-HRD1 ERAD substrates. Furthermore, they found that SEL1L-HRD1 ERAD suppressed the biosynthesis of glycosylphosphatidylinositol (GPI)-anchored proteins through controlling the abundance of GPI-transamidase via its catalytic subunit PIGK, and PIGK disease mutants were unstable and quickly degraded by SEL1L-HRD1 ERAD. Overall, the results are very interesting, and this work contains valuable information for advancing our understanding of ERAD.

“Following the development of a unique label-free proteomics screen...,” and “Here we have developed a unique, label-free, immunoprecipitation (IP)-based proteomic screening strategy to...” The label-free proteomics method itself is not new in this work, except that they generated several SEL1L-specific antibodies and found that one of them was suitable for IP of endogenous SEL1L protein in mammalian cells. Therefore, it might be better for not emphasizing the method development. This does not affect the novelty of this work.

The overlaps among three independent experiments are relatively low. For example, in Fig. 3a, only 22 proteins were identified in all three experiments while 180 proteins were exclusively identified in one experiment. It might be better to give some explanations. One minor point: the labels in Fig. 3a should be Expt. 1, 2, or 3.

“PLA2G12A and MLEC were accumulated in both HEK293T cells and BAT when SEL1L or HRD1 were deficient (Fig. 5b), in a transcription independent manner (Extended Data Fig. 4a). Degradation of PLA2G12A and MLEC in HEK293T cells was indeed SEL1L- and HRD1-dependent as both proteins was significantly stabilized in the absence of SEL1L-HRD1 ERAD.” In Fig. 5b, it is very clear that PLA2G12A and MLEC were accumulated when SEL1L or HRD1 was deficient. In Fig. 5c, both proteins were significantly stabilized in the absence of SEL1L-HRD1 ERAD. However, in WT, these two proteins (especially

PLA2G12A) also seemed to be relatively stable, which is quite different from FUCA2 in Fig. 5f. It might be better to explain this. Regarding Fig. 5, another minor point: in Fig. 5b, there is one band for MLEC, while two bands appear in Fig. 5c. Why?

Minor points:

In Figure 2, move “e” upward to the level of IgG. Otherwise, the labels in the box look like to belong to “b”.

In Supplementary Info file, “Cysteines were reduced by adding 50 μ l of 10 mM DTT...” Actually, the disulfide bonds in proteins are reduced. It might also be fine to write: “Proteins were reduced by...”

In Supporting Tables (Excel file), please pay attention to significant figures.

Reviewer #3 (Remarks to the Author):

Overview

In this manuscript, the authors present a new strategy to discover and identify endogenous substrates of the Hrd1-SEL1L ubiquitin ligase complex and ERAD, applying it to both in vitro (cell lines) and in vivo (mouse adipose tissue). They first characterise a SEL1L antibody able to isolate Hrd-containing complexes with high efficiency – demonstrating its selectivity by performing comparative SILAC on WT and SEL1L-/- lines (Fig 1). The authors then apply this strategy first to HEK293 cells (Fig. 2) and subsequently brown adipose tissue (BAT, Fig 3) where they uncover proteins that accumulate with SEL1L in the absence of Hrd1. Importantly, comparative analysis of RNASeq data for WT and Hrd-/- revealed little to no difference. The authors then go on to reflect on the shared and unique ERAD substrates present in these cells of different origins (Fig. 4), highlighting the enrichment of mitochondrial, lipid, redox and secretion related proteins in BATs and UPR, nuclear and lysosomal function in HEK293s (Fig. 4) and suggesting a set of cell-type specific substrates for this ERAD pathway. Next, the authors sought to validate substrates from their MS data directly, demonstrating elevated levels of substrates such as PLA2G12A, MLEC, FUCA2, LPL and ADIPOQ in both Hrd1-/- and SEL1L-/- and some stabilisation during translational arrest (Fig. 5). They then focus in on the GPI transferase subunit PIGK as a substrate and provide evidence that its association with SEL1L is greater in a Hrd1-/- and exhibits less polyubiquitylation and stabilisation of its turnover kinetics (Fig. 6). The consequence of accumulating the GPI-related enzyme in the absence of SEL1L-HRD1 mediated ERAD was pursued in Figure 7, where the authors report an increase in GPI-linked proteins at the cell surface. The removal of SEL1L is masked by loss of PIGK, indicating that the effect on GPI0-linked accumulation is due to PIGK levels and not directly via SEL1L. Reintroduction of either SEL1L

or HRD1 reverses the increase in GPI-linked cell surface proteins (Fig. 7g,h), indicating a role for them in GPI-anchored protein biosynthesis. Finally, 5 different PIGK disease mutants were shown to be stabilised in HRD^{-/-} cells, forming HMW aggregates due to aberrant intramolecular disulphide bonds and suggesting that ERAD serves to limit pathogenicity of the disease.

Overall impression

The identification of physiological substrates of ERAD is of great interest not only to the ER community but also represents an important advance in ubiquitin biology in general. It should be of general interest both in its strategy and in its findings. This manuscript does a very good job of executing this analysis and highlighting the differences that are undoubtedly present between different tissues, highlighting the general and specific functions ERAD plays. Overall, the manuscript is well written, the experiments performed to a high standard, the data of good quality, and the appropriate statistical analyses have mostly been included where necessary and are described. The Methods and Materials describe the experiments that have been performed. Some concern might arise from the fact that this analysis is unlikely to contain any of the cytoplasmic substrates that are regulated by HRD1 degradation but as much headway is made here for lumenal and membrane proteins, this limitation does not compromise the overall work. There was some concern with the number of figures in the paper and some issues that arose with consistency among western blots and the band intensities within them (detailed below) which should be addressed. The manuscript is an advancement of understanding of the physiological substrates targeted for ERAD via SEL1L-Hrd1. The comments/suggestions that this reviewer has regarding the data presented are listed below.

Queries & Comments

1. In Figure 1a, ProAVPG57S levels increase in lysate from either Hrd1 or SEL1L KO but that is not reflected in the pulldown from Hrd1^{-/-}. Could the authors explain why this might be. Also, expression of ProAVPG57S increases the steady-state levels of both Hrd1 and SEL1L. Does its expression elicit a UPR?
2. Figure 4 could be moved to Extended data as it does not contribute much to the interpretation of the data
3. Referring to Figure 5C, the authors state that “Degradation of PLA2G12A and MLEC in HEK293T cells was indeed SEL1L- and HRD-1 dependent as both proteins was (should be were here) significantly stabilized in the absence of SEL1L-HRD1 ERAD”, however PLA2G12A does not appear to be being turned over at all in WT cells upon addition of CHX, which might suggest that PLA2G12A is not a very strong/good substrate. Has the t_{1/2} of PLA2G12A been established? Could the authors explain why no/little turnover is observed in their western blot? Perhaps these values could be quantified to better

reflect the turnover. Why is the 0 hr of SEL1L (lane 1) roughly as intense as that of WT (lane 4) when in Fig 5B, there is a 2.6x difference (compare lanes 1,2 and 3,4)?

4. In Figure 6c, what is the nature of the second (upper) band displayed in the PIGT western blot which is present in Cortex but not Cerebellar tissue in the absence of Tamoxifen and vice versa for WT and + Tamoxifen? This is not clear. Could it be that the lanes for Tamoxifen (-) have been inadvertently switched and misloaded for Cortex and Cerebellum?

5. In Figure 6d, the input levels of PIGK do not appear that different in the lysate of WT and SEL1L^{-/-} (compare to Figure 6b). Perhaps this might be quantified to better demonstrate that that starting amounts are different.

6. In Figure 6d, why does PIGT appear to be enriched in a SEL1L IP but is not found in the MS? Why does it not enrich when PIGK is stabilised? Moreover, in Figure 6e, why doesn't the relative amount of PIGT brought down by PIGK IP change even though PIGK levels change between WT and SEL1L or HRD1 KO? Do these represent assembled and unassembled forms of PIGK?

7. In Figure 6d, why is the SEL1L band from a SEL1L IP of a WT sample so poor (compare to Fig 1b)?

8. For Figure 6f, could the authors include treatment with a recombinant DUB to demonstrate direct modification of PIGK by ubiquitin. Prior to IP, were these lysates denatured to ensure that coprecipitating factors are not the source of Ub immunoreactivity? Could the authors speculate on the origins of PIGK ubiquitylation that remains in the absence of Hrd1?

9. In Figure 6f, could the authors explain why the Hrd1 input band in Figure 6f is reduced when MG132 is added?

We thank all reviewers for their insightful and constructive comments! We now have carefully addressed all the comments from the reviewers, which have been instrumental to further improve and strengthen our manuscript.

Reviewer #1 (Remarks to the Author):

Authors developed a new proteomic screen for endogenous substrates of SEL1L-HRD1 ERAD useful for both cultured cells and tissues/organs. The developed method was applied to HEK293T cells and brown adipose tissue from mice, and was proven to be highly efficient in picking up candidates of SEL1L-HRD1 ERAD substrates. In fact, more than 100 candidates were obtained from both sources and ~20 % were common and the rests were cell-type specific. Five selected candidates from the common type were experimentally validated using SEL1L-KO and HRD1-KO HEK293T cells and/or Sel1L-defective mice, confirming usefulness of the developed method. The substrate candidates of known functions are involved in various cellular pathways including GPI-anchored protein biogenesis. Authors then focused on PIGK, the catalytic component of GPI transamidase complex that mediates attachment of GPI to precursor proteins, and demonstrated PIGK is a substrate of SEL1L-HRD1 ERAD. Based on a series of experiments, authors conclude that SEL1L-HRD1 ERAD suppresses biogenesis of GPI-anchored proteins by targeting PIGK for proteasomal degradation. Application of the developed method to various tissues/organs should be useful to determine endogenous SEL1L-HRD1 ERAD substrates specific to respective tissues/organs. Concerning the part of SEL1L-HRD1 ERAD effects on biogenesis of GPI-anchored proteins, conclusions are not fully supported by the provided results. I have following points to be addressed by authors.

Major points:

1. It is clearly shown in Fig 6 that PIGK protein levels increased under SEL1-HRD1 ERAD defective conditions in HEK293T cells, and mouse brown adipose tissue, cerebral cortex and cerebellum. In contrast, PIGT protein levels did not increase after SEL1L/Sel1L-KO or HRD1-KO. When PIGK was immunoprecipitated from SEL1L-KO, HRD1-KO and WT HEK293T cells, co-precipitated PIGT levels seem to be similar in three cells (Fig 6e). These results suggest that levels of GPI transamidase complex containing PIGK and PIGT did not increase under SEL1-HRD1 ERAD defective conditions and that PIGK free from GPI transamidase increased. It is likely that PIGK is synthesized in excess relative to PIGT in WT cells and excessive fraction of PIGK is degraded by SEL1-HRD1 ERAD.

We thank this Reviewer’s insightful comments. Our new data now showed that immunoprecipitation of PIGK in HRD1 KO cells did pull down 30% more PIGT than that in WT cells (**Response Figure 1a**). Hence, these data suggested that the GPI transamidase complex was elevated in ERAD deficient cells. Based on these new figures, we propose that PIGK is the limiting factor in the complex, whose stability is controlled by SEL1L-HRD1 ERAD, and that the abundance of both PIGK and transamidase complex is increased in the absence of SEL1L-HRD1 ERAD (**Response Figure 1b**). We now have revised the Results and Discussion section as pasted below.

Response Figure 1. Enhanced formation of the PIGK-PIGT complex in the absence of ERAD (a) and the model (b). a, Representative Western blot analysis following IP of endogenous PIGK in SEL1L KO, WT and HRD1 KO HEK293T cells showing the interactions between PIGK and SEL1L or PIGT. * P < 0.05 using one-way ANOVA followed by Dunnett's multiple comparisons test. b, Proposed model for the role of SEL1L-HRD1 ERAD in the biogenesis of GPI-APs in the ER. Misfolded PIGK protein is degraded by SEL1L-HRD1 ERAD, limiting the abundance and function of GPI-transamidase complex. In the absence of SEL1L-HRD1 ERAD, PIGK accumulated, leading to increased abundance of GPI-transamidase complex and consequently elevated production of GPI-APs.

Results: “We then asked how SEL1L-HRD1 ERAD recognizes PIGK. In the SEL1L-IP-MS proteomic screens, other subunits of the GPI-transamidase complex including PIGT, PIGS, PIGU and GPA1 were either not identified as high confidence hits or not immunoprecipitated by SEL1L (Extended Data Fig. 6a). Unlike PIGK, PIGT was unchanged after acute SEL1L deletion in the cortex and cerebellum of inducible Sel1L-deficient mouse model (Fig. 6c). In addition, in SEL1L-IP of HEK293T cells, PIGT-SEL1L interaction was not increased in the absence of HRD1, unlike that of PIGK-SEL1L (Fig. 6d). Lastly, unlike PIGK, PIGT and three other subunits were not stabilized in HRD1 KO HEK293T cells (Fig. 6g and Extended Data Fig. 6b). Therefore, we concluded that SEL1L-HRD1 ERAD specifically targets PIGK protein for proteasomal degradation.”

Discussion: “We showed that SEL1L-HRD1 ERAD targets PIGK for proteasomal degradation, thereby negatively regulating the abundance of functional GPI-transamidase complex and hence the biogenesis of GPI-anchored proteins in the ER. Intriguingly, this effect of SEL1L-HRD1 ERAD on the activity of the GPI-transamidase complex seems specifically mediated through PIGK, not the other subunits (Fig. 8e).”

2. Both cell surface levels of CD59, a GPI-AP, and staining levels by FLAER, a fluorescent probe for GPI moiety in some GPI-APs, increased in SEL1L-KO and HRD1-KO HEK293T cells. Increase was modest but was dependent upon SEL1L- and HRD1-KO because the increased CD59 levels turned to WT levels after rescue by corresponding cDNA (Fig 7). Because data in Fig 6 did not support an increase of GPI transamidase, a possible reason for the increased cell surface levels of CD59 and FLAER staining might be related to stressed ER conditions in SEL1L-HRD1 ERAD defective cells. It is not surprising if trafficking/turnover of CD59 is modified in SEL1L-HRD1 ERAD defective cells, resulting in modest increase in the cell surface level. Without showing increase of GPI transamidase complex or increased generation of GPI-anchored CD59 in the ER, the conclusion that SEL1L-HRD1 ERAD suppresses biosynthesis of GPI-APs via PIGK should be modified.

We thank this Reviewer’s insightful comments. Our new data shown in **Response Figure 1** supports the notion that the GPI transamidase complex was increased, albeit mildly, in ERAD-deficient cells. Our new data further show that treatment with ER stress inducer thapsigargin (Tg) triggered strong ER stress but failed to increase CD59 level at the cell surface (**Response Figure 2a-b**), and that SEL1L- and HRD1-deficiency only caused subtle ER stress, compared to Tg-treated cells (**Response Figure 2b**).

Furthermore, following the treatment of a phosphatidylinositol (PI)-specific phospholipase C (PI-PLC) which cleaves and releases surface GPI-APs¹, our new data revealed elevated intracellular CD59 in ERAD KO cells compared to WT cells using both flow cytometric and Western blot analyses (**Response Figure 2c-d**), providing additional (indirect) evidence for elevated GPI transamidase activity. Lastly, we

have modified our conclusion to “SEL1L-HRD1 ERAD attenuates the biosynthesis of GPI-APs, at least in part, via PIGK”.

Response Fig. 2. ERAD effect on GPI-AP production is uncoupled from ER stress and increased intracellular levels of CD59 proteins in ERAD-deficient cells.

Response Fig. 2. ERAD effect on GPI-AP production is uncoupled from ER stress and increased intracellular levels of CD59 proteins in ERAD-deficient cells. **a**, Quantitation for flow cytometric analysis of surface CD59 protein in HEK293T cells treated with or without 50 nM thapsigargin (Tg) for 4 hours (n=3 independent repeats). **b**, RT-PCR analysis of XBP1 splicing in HEK293T cells. u/s, unspliced/spliced Xbp1. Quantitation of the relative abundance of spliced XBP1 bands was indicated below. **c**, Quantitation of the flow cytometric analysis of intracellular CD59 in HEK293T cells treated with 5 U/mL phosphatidylinositol-specific phospholipase C (PI-PLC) to deplete surface GPI-anchored proteins (n=3 independent repeats). **d**, Representative immunoblotting of intracellular CD59 in HEK293T cells treated with PI-PLC

(n=3 independent repeats). Values represent mean \pm SEM. n.s., not significant, * $P < 0.05$, ** $P < 0.01$, *** $P < 0.001$ using one-way ANOVA followed by Dunnett's multiple comparisons test.

Other points:

1. It is unusual to use -/- for KO in HEK293T cells. Symbols +/+, +/- and -/- are used when cells/organisms are diploid where -/- means both alleles are dead. HEK293T cells are generally triploid. Only when SEL1L and HRD1 genes in HEK293T are exceptionally diploid and biallelic mutations are shown, -/- is appropriate. Authors showed complete loss of SEL1L and HRD1 proteins after KO, suggesting successful KO of all alleles. I suggest to use SEL1L-KO instead of SEL1L-/-.

We have updated the labels as suggested throughout the text.

2. Lines 104-105. This statement does not fit with data shown in Fig 6. If SEL1L-HRD1- ERAD is involved only in degradation of misfolded PIGK, how it plays a key role in GPI-AP biogenesis.

We thank the reviewer for this great point. We now have revised the statement in the Introduction to "Following the validation of several hits, we further show that SEL1L-HRD1 ERAD attenuates the biogenesis of GPI-anchored proteins, at least in part, by targeting PIGK protein for proteasomal degradation."

3. Fig 3e and related file.

CLPTM1 is listed as a factor in GPI-AP biogenesis. As far as I know, it has never been shown.

In a haploid genetic screen for factors required for efficient GPI-inositol diacylation, CLPTM1 was identified and CLPTM1 deficiency conferred partial resistance to phosphatidylinositol (PI)-specific phospholipase C in HEK293T cells¹. These data support its possible involvement in biogenesis of GPI-APs; however, as pointed out by the reviewer that its role in GPI-AP biogenesis remains unclear. We now have taken it out in the revised manuscript.

4. Line 271, "GPI-transamidase complex catalyzes the last step of biosynthesis of GPI anchored proteins" is not accurate. The enzyme generates nascent GPI-anchored proteins, whose GPI-anchor moiety is structurally immature and undergoes several maturation reactions in later steps.

We thank the reviewer for this great point. We have rephrased the statement to "PIGK is the catalytic subunit of the ER-resident GPI-transamidase complex, which catalyzes the generation of nascent GPI-anchored proteins".

5. Fig 6d. In IP: SEL1L part, SEL1L band in WT cell sample is very faint. PIGK is faint whereas PIGT is stronger than one in HRD1-KO. Please explain these unexpected profiles.

We thank the reviewer for this great point. We now have repeated this experiment and new Fig. 6d is now included in the revised manuscript, which shows strong SEL1L and PIGK signals and similar PIGT signals in WT vs. HRD1 KO cells (**Response Figure 3**). This data demonstrates that SEL1L-HRD1 ERAD interacts with PIGK, but not PIGT.

Response Fig. 3. PIGK is an endogenous SEL1L-HRD1 ERAD substrate.

Representative Western blot analysis following IP of endogenous SEL1L in HEK293T cells with quantitation shown below the blots (n=5 independent repeats). n.s., not significant, ** $P < 0.01$ using one-way ANOVA followed by Dunnett's multiple comparisons test.

6. Fig 6f. When PIGK gets polyUb, PIGK should be seen as smear in the presence of MG132. Why only intact PIGK is shown?

We thank the reviewer for this great point. We have repeated this experiment but do not see the smear PIGK (**Response Figure 4**). The reason may be the percent of Ubiquitinated PIGK protein below the detection limit.

Response Fig. 4. PIGK is polyubiquitinated by HRD1. Immunoblot analyses of polyubiquitination following denaturing immunoprecipitation of PIGK-FLAG in WT and *HRD1* KO HEK293T cells treated with or without MG132 treatment for 2 hours (n=2 independent repeats).

7. Lines 295-302. Authors asked whether SEL1-HRD1 ERAD recognizes PIGK in a complex or as an individual protein. An answer to this question is not clearly stated in this paragraph.

Thank you for pointing this out. We now have added a conclusion at the end of this paragraph. *“Therefore, we concluded that SEL1L-HRD1 ERAD specifically targets PIGK protein for proteasomal degradation.”*

8. M & M. Section “Immunoprecipitation. A lysis buffer contains 0.2% NP40 and 0.1% Triton-X100. These two detergents are similar. Is there any specific reason for mixing them?”

We used the combination of detergents for the following reasons: its effectiveness in pulling down the entire ERAD complex as previously demonstrated² and in the immunoprecipitation of the substrate-ERAD interaction in HEK293T cells as previously demonstrated³.

Reviewer #2 (Remarks to the Author):

In this work, the authors employed label-free MS-based proteomics to identify the substrates of Endoplasmic Reticulum-Associated Degradation (ERAD). In the SEL1L-HRD1 complex, which represents the most conserved branch of ERAD, SEL1L serves as a scaffolding protein for the ER-resident E3 ubiquitin ligase and retrotranslocon HRD1, and it recruits substrates. Using an antibody against SEL1L, they comprehensively identified the interactors of SEL1L. Combining a computational method for stringent filtering, more than 100 potential substrates were identified in both the human kidney cell line (HEK293T) and the mouse brown adipose tissue, respectively. They further validated several potential substrates, i.e., PLA2G12A, MLEC, FUCA2, ADIPOQ and LPL, as endogenous SEL1L-HRD1 ERAD substrates. Furthermore, they found that SEL1L-HRD1 ERAD suppressed the biosynthesis of glycosylphosphatidylinositol (GPI)-anchored proteins through controlling the abundance of GPI-transamidase via its catalytic subunit PIGK, and PIGK disease mutants were unstable and quickly degraded by SEL1L-HRD1 ERAD. Overall, the results are very interesting, and this work contains valuable information for advancing our understanding of ERAD.

“Following the development of a unique label-free proteomics screen...,” and “Here we have developed a unique, label-free, immunoprecipitation (IP)-based proteomic screening strategy to...” The label-free proteomics method itself is not new in this work, except that they generated several SEL1L-specific antibodies and found that one of them was suitable for IP of endogenous SEL1L protein in mammalian cells. Therefore, it might be better for not emphasizing the method development. This does not affect the novelty of this work.

We agree with the reviewer on this great point, have rephrased these statements by deleting “unique” as suggested and de-emphasized the methods development throughout the manuscript.

The overlaps among three independent experiments are relatively low. For example, in Fig. 3a, only 22 proteins were identified in all three experiments while 180 proteins were exclusively identified in one experiment. It might be better to give some explanations. One minor point: the labels in Fig. 3a should be Expt. 1, 2, or 3.

We thank the Reviewer for this great point. To increase the rigor of the study and to identify adipocyte-specific candidates, we performed the experiments using two different cell types of adipocytes: one brown adipose tissue (BAT) and the other “differentiated white adipocytes”. Moreover, 4 mice were used

in BAT Expt. 1 vs. 1 mouse in BAT Expt. 2 – hence the overall signals in Expt. 1 were much more than those in Expt. 2. We now have added following statements in the figure legends and methods to clarify the confusion. The labels in Fig. 2b and Fig. 3a have been updated to Expt.1, 2, or 3.

Figure Legends: “**Figure 3. Proteomic screening for SEL1L-HRD1 ERAD substrates in brown adipocytes.** **a**, Venn diagram showing the overlaps of substrate candidates identified from three independent experiments including two experiments from brown adipose tissues (BAT, n=4 mice in Expt. 1 and 1 mouse in Expt. 2) and one experiment from differentiated white adipocytes.”

Methods: “HEK293T cells or differentiated white adipocytes were collected and pooled from 4x 10 cm culture dishes. IP in BAT was performed using frozen BAT from 12-week-old mice (n=4 in Expt. 1; and n=1 in Expt. 2). Cells or tissues were lysed with cold IP lysis buffer...”

“PLA2G12A and MLEC were accumulated in both HEK293T cells and BAT when SEL1L or HRD1 were deficient (Fig. 5b), in a transcription independent manner (Extended Data Fig. 4a). Degradation of PLA2G12A and MLEC in HEK293T cells was indeed SEL1L- and HRD1-dependent as both proteins was significantly stabilized in the absence of SEL1L-HRD1 ERAD.” In Fig. 5b, it is very clear that PLA2G12A and MLEC were accumulated when SEL1L or HRD1 was deficient. In Fig. 5c, both proteins were significantly stabilized in the absence of SEL1L-HRD1 ERAD. However, in WT, these two proteins (especially PLA2G12A) also seemed to be relatively stable, which is quite different from FUCA2 in Fig. 5f. It might be better to explain this. Regarding Fig. 5, another minor point: in Fig. 5b, there is one band for MLEC, while two bands appear in Fig. 5c. Why?”

We thank the reviewer for this great point. Our data in Fig 5 showed that different substrates may have different half-lives in WT cells, with some (e.g. PLA2G12A and MLEC) more stable than the others (e.g. FUCA2). This may be related to the abundance and/or intrinsic biochemical and biophysical properties of

the substrates. Our new data consistently showed that MLEC runs as two bands. We now have updated the PLA2G12A and MLEC blots in the original Fig 5b-c with the ones shown in **Response Fig. 5a-b**. Quantitation of protein half-lives are now shown in Figure 5d and 5h (**Response Fig. 5c**).

Response Fig. 5. Validation of a subset of ERAD substrates. **a**, immunoblot analysis showing ERAD-dependent degradation of PLA2G12A and MLEC proteins in HEK293T cells following cycloheximide (CHX) treatment for the indicated times. **b**, Immunoblot analysis of PLA2G12A and MLEC in HEK293T cells and BAT (n=4-6 samples from 2 independent experiments). **c**, Quantitation of protein half-life in HEK293T cells from 3 independent experiments. * P < 0.05, ** P < 0.01 using one-way ANOVA followed by Dunnett’s multiple comparisons test (for HEK293T) and Student’s t test (for BAT).

Minor points:

In Figure 2, move “e” upward to the level of IgG. Otherwise, the labels in the box look like to belong to “b”.

In Supplementary Info file, “Cysteines were reduced by adding 50 µl of 10 mM DTT...” Actually, the disulfide bonds in proteins are reduced. It might also be fine to write: “Proteins were reduced by...”

In Supporting Tables (Excel file), please pay attention to significant figures.

We thank the reviewer for pointing out these issues. We now have made changes as suggested.

Reviewer #3 (Remarks to the Author):

In this manuscript, the authors present a new strategy to discover and identify endogenous substrates of the Hrd1-SEL1L ubiquitin ligase complex and ERAD, applying it to both in vitro (cell lines) and in vivo (mouse adipose tissue). They first characterise a SEL1L antibody able to isolate Hrd-containing complexes with high efficiency – demonstrating its selectivity by performing comparative SILAC on WT and SEL1L^{-/-} lines (Fig 1). The authors then apply this strategy first to HEK293 cells (Fig. 2) and subsequently brown adipose tissue (BAT, Fig 3) where they uncover proteins that accumulate with SEL1L in the absence of Hrd1. Importantly, comparative analysis of RNASeq data for WT and Hrd^{-/-} revealed little to no difference. The authors then go on to reflect on the shared and unique ERAD substrates present in these cells of different origins (Fig. 4), highlighting the enrichment of mitochondrial, lipid, redox and secretion related proteins in BATs and UPR, nuclear and lysosomal function in HEK293s (Fig. 4) and suggesting a set of cell-type specific substrates for this ERAD pathway. Next, the authors sought to validate substrates from their MS data directly, demonstrating elevated levels of substrates such as PLA2G12A, MLEC, FUCA2, LPL and ADIPOQ in both Hrd1^{-/-} and SEL1L^{-/-} and some stabilisation during translational arrest (Fig. 5). They then focus in on the GPI transferase subunit PIGK as a substrate and provide evidence that its association with SEL1L is greater in a Hrd1^{-/-} and exhibits less polyubiquitylation and stabilisation of its turnover kinetics (Fig. 6). The consequence of accumulating the GPI-related enzyme in the absence of SEL1L-HRD1 mediated ERAD was pursued in Figure 7, where the authors report an increase in GPI-linked proteins at the cell surface. The removal of SEL1L is masked by loss of PIGK, indicating that the effect on GPI-linked accumulation is due to PIGK levels and not directly via SEL1L. Reintroduction of either SEL1L or HRD1 reverses the increase in GPI-linked cell surface proteins (Fig. 7g,h), indicating a role for them in GPI-anchored protein biosynthesis. Finally, 5 different PIGK disease mutants were shown to be stabilised in HRD^{-/-} cells, forming HMW aggregates due to aberrant intramolecular disulphide bonds and suggesting that ERAD serves to limit pathogenicity of the disease.

Overall impression

The identification of physiological substrates of ERAD is of great interest not only to the ER community but also represents an important advance in ubiquitin biology in general. It should be of general interest both in its strategy and in its findings. This manuscript does a very good job of executing this analysis and highlighting the differences that are undoubtedly present between different tissues, highlighting the general and specific functions ERAD plays. Overall, the manuscript is well written, the experiments performed to a high standard, the data of good quality, and the appropriate statistical analyses have mostly been included where necessary and are described. The Methods and Materials describe the experiments that have been performed. Some concern might arise from the fact that this analysis is unlikely to contain any of the cytoplasmic substrates that are regulated by HRD1 degradation but as much headway is made here for luminal and membrane proteins, this limitation does not compromise the overall work. There was some concern with the number of figures in the paper and some issues that arose with consistency among western blots and the band intensities within them (detailed below) which should be addressed. The manuscript is an advancement of understanding of the physiological substrates targeted for ERAD via SEL1L-Hrd1. The comments/suggestions that this reviewer has regarding the data presented are listed below.

We thank and agree with the reviewer for the great points. The experimental design of SEL1L IP, which has short cytosolic tail, does limit our ability to identify cytosolic substrates. We now have added a statement to this potential caveat at the end of the Discussion.

Discussion: *“Taken together, while we acknowledge that a caveat of our proteomic screening is its limited ability to identify HRD1 cytosolic substrates, further studies are required to delineate the importance of SEL1L-HRD1 ERAD in the quality control of proteins in other organelles and its underlying mechanism(s).”*

Queries & Comments:

1. In Figure 1a, ProAVPG57S levels increase in lysate from either Hrd1 or SEL1L KO but that is not reflected in the pull-down from Hrd1^{-/-}. Could the authors explain why this might be. Also, expression of ProAVPG57S increases the steady-state levels of both Hrd1 and SEL1L. Does its expression elicit a UPR?

We thank the reviewer for this important comment. We now have repeated this experiment and confirmed that more ProAVP^{G57S} was pulled down in SEL1L or HRD1 KO than in WT cells. In addition, we have not consistently seen the increase of SEL1L and HRD1 protein levels in cells expressing ProAVP^{G57S}. A new experiment is now shown in Fig. 1a (**Response Fig. 6**).

Response Fig. 6. The interaction between SEL1L and an ERAD substrate HA-tagged ProAVP^{G57S}. Immunoprecipitation of HA-ProAVP^{G57S} in WT, SEL1L and HRD1 KO HEK293T cells transfected with ProAVP^{G57S}-HA, showing the accumulation of the substrates and increased interaction between the substrate and SEL1L in HRD1 KO cells. Asterisk indicates a non-specific cross-reactive band.

2. Figure 4 could be moved to Extended data as it does not contribute much to the interpretation of the data.

We thank the reviewer for the suggestion. We'd like to keep Fig 4 in the main figure as it not only directly shows that ERAD regulates various cellular processes in a cell type-specific manner, but helps transition to subsequent Figures. If the reviewer deems this figure absolutely unhelpful in the main figure, we will be happy to move it to the Extended data during the next revision.

3. Referring to Figure 5C, the authors state that "Degradation of PLA2G12A and MLEC in HEK293T cells was indeed SEL1L- and HRD-1 dependent as both proteins was (should be were here) significantly stabilized in the absence of SEI1L-HRD1 ERAD", however PLA2G12A does not appear to be being turned over at all in WT cells upon addition of CHX, which might suggest that PLA2G12A is not a very strong/good substrate. Has the t_{1/2} of PLA2G12A been established? Could the authors explain why no/little turnover is observed in their western blot? Perhaps these values could be quantified to better reflect the turnover. Why is the 0 hr of SEL1L (lane 1) roughly as intense as that of WT (lane 4) when in Fig 5B, there is a 2.6x difference (compare lanes 1,2 and 3,4)?

We thank the Reviewer for this great point. We now updated the original Fig 5c with the new data and quantitation for the CHX chase experiments in Fig 5d (**Response Fig. 5**).

4. In Figure 6c, what is the nature of the second (upper) band displayed in the PIGT western blot which is present in Cortex but not Cerebellar tissue in the absence of Tamoxifen and vice versa for WT and + Tamoxifen? This is not clear. Could it be that the lanes for Tamoxifen (-) have been inadvertently switched and misloaded for Cortex and Cerebellum?

We thank the Reviewer for this great point. We don't know the nature of the additional faint band in PIGT blots in tissues as it is not present in cells – maybe non-specific. Our new experiment also showed that the upper band was not unique to tamoxifen-treated samples (**Response Figure 7**). PIGT/PIGK/SEL1L

blots were performed together so we are sure that the samples were not mixed up or misloaded.

Response Fig. 7. Comparison of PIGT bands in tissues vs. cells. Immunoblot analyses of endogenous PIGT in the cerebral cortex of WT and Sel1L^{ERCre} mice with or without 100 mg/kg tamoxifen injection. Last two lanes are WT and PIGT KO HEK293T as controls.

5. In Figure 6d, the input levels of PIGK do not appear that different in the lysate of WT and SEL1L^{-/-} (compare to Figure 6b). Perhaps this might be quantified to better demonstrate that that starting amounts are different.

We thank the reviewer for this great point. We now have included new data with quantitation in Fig 6d (**Response Figure 8**).

Response Fig. 8. Quantitation of SEL1L interaction with PIGK and PIGT. Representative Western blot analysis following IP of endogenous SEL1L in HEK293T cells with quantitation shown below the blots. (n=5 independent repeats). n.s., not significant, ** $P < 0.01$ using one-way ANOVA followed by Dunnett's multiple comparisons test.

6. In Figure 6d, why does PIGT appear to be enriched in a SEL1L IP but is not found in the MS? Why does it not enrich when PIGK is stabilised? Moreover, in Figure 6e, why doesn't the relative amount of PIGT brought down by PIGK IP change even though PIGK levels change between WT and SEL1L or HRD1 KOs? Do these represent assembled and unassembled forms of PIGK?

We thank the reviewer for this great point. PIGT was present in IP-MS, but it was not enriched in *HRD1 KO* cells (Extended Data Fig. 6a). Our new data showed that PIGT is not an ERAD substrate (**Response Figure 9a**). In addition, we have repeated the PIGK IP experiment and following quantitation, we observed a 30% increase of PIGT pulled down by PIGK in HRD1 KO cells compared to WT cells, which is now shown in Fig 6e (**Response Figure 9b**). This data supported the model of elevated transamidase complex in ERAD deficient cells (Figure 8e).

increase of PIGT pulled down by PIGK in HRD1 KO cells compared to WT cells, which is now shown in Fig 6e (**Response Figure 9b**). This data supported the model of elevated transamidase complex in ERAD deficient cells (Figure 8e).

Response Fig. 9. Elevated PIGK-PIGT interaction in the absence of ERAD. **a**, Immunoblot analyses of endogenous PIGK or PIGT protein in HEK293T cells treated with 50 µg/ml cycloheximide (CHX) for the indicated times with quantitation shown below (n=5/3 independent repeats for PIGK/PIGT). **b**, Representative Western blot analysis following IP of endogenous PIGK in HEK293T cells showing the interactions between PIGK and SEL1L or PIGT. n.s., not significant, * $P < 0.05$, *** $P < 0.001$ using one-way ANOVA followed by Dunnett's multiple comparisons test.

7. In Figure 6d, why is the SEL1L band from a SEL1L IP of a WT sample so poor (compare to Fig 1b)? We thank the reviewer for this great point. We now have repeated the experiment and a new Fig. 6d is now included with strong SEL1L signals in WT cells (**Response Figure 8**).

8. For Figure 6f, could the authors include treatment with a recombinant DUB to demonstrate direct modification of PIGK by ubiquitin. Prior to IP, were these lysates denatured to ensure that coprecipitating factors are not the source of Ub immunoreactivity? Could the authors speculate on the origins of PIGK ubiquitylation that remains in the absence of Hrd1? We thank the reviewer for this great point. We now have performed denaturing PIGK IP followed by DUB treatment with ubiquitin specific protease (USP) 2, which showed that the PIGK is indeed directly ubiquitinated in an HRD1 dependent manner (**Response Figure 10**). Other ERAD branches or the residual HRD1 activity may account for the remaining PIGK ubiquitination in HRD1 KO cells.

Response Fig. 10. PIGK is polyubiquitinated by HRD1. Immunoblot analyses of polyubiquitination following denaturing IP of PIGK-FLAG in transfected WT and *HRD1 KO* HEK293T cells treated with or without MG132 treatment for 2 hours (n=2 independent repeats).

9. In Figure 6f, could the authors explain why the Hrd1 input band in Figure 6f is reduced when MG132 is added?

We thank the reviewer for this important point. We do not consistently see reduced HRD1 protein levels in samples treated with MG132 as shown in **Response Figure 10**.

References:

1. Liu, Y.S. *et al.* N-Glycan-dependent protein folding and endoplasmic reticulum retention regulate GPI-anchor processing. *J Cell Biol* **217**, 585-599 (2018).
2. Lin, L. *et al.* PAWH1 and PAWH2 are plant-specific components of an Arabidopsis endoplasmic reticulum-associated degradation complex. *Nat Commun* **10**, 3492 (2019).
3. Zhou, Z. *et al.* Endoplasmic reticulum-associated degradation regulates mitochondrial dynamics in brown adipocytes. *Science* **368**, 54-60 (2020).

REVIEWERS' COMMENTS

Reviewer #1 (Remarks to the Author):

Authors addressed all my points and revised the manuscript by providing new data and/or modified sentences appropriately.

I suggest a small update. As more detailed 3D structure of GPI transamidase was recently reported (Xu Y et al, Nat Commun, 2023, 14:5520), it would be useful for readers if schematic diagram shown in Expanded Data Fig 5a is updated based on new data.

Reviewer #2 (Remarks to the Author):

I think that the authors addressed my comments well. Now the revised manuscript has a better quality for being published in Nat. Comm.

Reviewer #3 (Remarks to the Author):

I am happy with the changes made by the authors in the revised manuscript. My concerns have been addressed.